# Derivation of Russian-specific reference intervals for complete blood count, iron markers and related vitamins

**Anna Ruzhanskaya**[1], **Kiyoshi Ichihara**[2]\*, **Elena Sukhacheva**[3], **Irina Skibo**[4], **Nina Vybornova**[4], **Dmitry Butlitski**[1], **Anton Vasiliev**[4], **Galina Agarkova**[1], **Ekaterina Vilenskaya**[1], **Vladimir Emanuel**[5], **Svetlana Lugovskaya**[6]

1 Beckman Coulter Inc., Moscow, Russia, 2 Faculty of Health Sciences, Yamaguchi University Graduate School of Medicine, Ube, Japan, 3 Beckman Coulter Eurocenter, Nyon, Switzerland, 4 Helix laboratory, Saint-Petersburg, Russia, 5 Pavlov First Saint-Petersburg State Medical University, Saint-Petersburg, Russia, 6 Russian Medical Academy of Continuing Professional Education of the Ministry of Health, Moscow, Russia

\* ichihara@yamaguchi-u.ac.jp

**Data Availability Statement:** We uploaded the source data as Supporting information in an Excel format.

## Abstract

### Objectives

This study aimed to establish reference intervals (RIs) for Russian adults for hematological parameters including related iron markers and vitamins. Sources of variation of reference values (RVs) and needs for secondary exclusion were explored for proper derivations of RIs.

### Methods

Following the harmonized protocol of the IFCC Committee on Reference Intervals and Decision Limits (C-RIDL), 506 healthy Russians (age 18−80; 46% male) were recruited. Complete blood counts (CBC) and leukocyte differentials, iron markers, vitamin B12, and folate were measured by Beckman Coulter's analyzers. Sources of variation were analyzed by multiple regression analysis, and ANOVAs, and the need for partitioning RVs was decided accordingly. Two schemes of excluding latent anemia were compared: (1) latent abnormal values exclusion method (LAVE) based on associations among CBC parameters, or (2) explicit exclusion of individuals with either ferritin or iron below the respective lower limit of the manufacturer. RIs were determined by the parametric method using two-parameter Box-Cox formula.

### Results

Gender-specific RIs were required for most analytes, while age-specific RIs were set only for ferritin in females. A BMI-related increase in RVs was prominently observed for reticulocyte parameters, hence we chose to exclude individuals with BMI>28 kg/m$^2$ when establishing the RIs. The LAVE method was more effective in excluding individuals with latent anemia, than exclusion based on low ferritin and/or iron values. International comparison revealed that Russian RIs featured a lower side shift of platelet counts. Similar to African

**Funding:** Beckman Coulter, LLC (Moscow, Russia; https://www.beckmancoulter.com/ru) supported the study of the assay reagents and Helix Laboratories Services (Saint-Petersburg, Russia; https://helix.ru/) assisted in the recruitment of volunteers, sample preparations, and provision of sampling equipment. The funders provided support in the form of salaries for authors (AR, ES, DB, GA, EV, IS, NV, AV), but did not take any role in the study design, data collection and analysis, decision to publish, or preparation of the manuscript. The specific roles of these authors are articulated in the 'author contributions' section.

**Competing interests:** The authors have declared that no competing interests exist.

**Abbreviations:** 2N-ANOVA, two-level-nested ANOVA; 3N-ANOVA, three-level-nested ANOVA; Mon, monocyte count; Mon%, percentage of monocyte count; Bas, basophil count; Bas%, percentage of basophil count; BC, Beckman Coulter; BMI, body mass index; BR, : bias ratio; CBC, complete blood count; CI, confidence interval; CLSI, Clinical and Laboratory Standards Institute; C-RIDL, Committee on Reference Intervals and Decision Limits; CV, coefficient of variation; Eos, eosinophil count; Eos%, percentage of eosinophil count; Fe, iron; Ferr, : ferritin; Hb, : hemoglobin; Ht, hematocrit; LL, low limit; IFU, instruction for use; IRF, immature reticulocyte fraction; LAVE, latent abnormal values exclusion; LHD%, percentage of low hemoglobin density; Lym, lymphocyte count; Lym%, percentage of lymphocyte count; M, male; MAF, microcytic anemia facto; MCV, mean corpuscular volume; MCH, mean corpuscular hemoglobin; MCHC, mean corpuscular hemoglobin concentration; Me, median; MPV, mean platelet count; MRA, multiple regression analysis; MRV, mean reticulocyte volume; Neu, neutrophil count; Neu%, percentage of neutrophil count; NP, non-parametric; P, parametric; PDW, platelet distribution width; PLT, platelet count; RBC, red blood cell count; RDW, red blood cell distribution width; Ret, reticulocyte count; Ret%, percentage of reticulocyte count; RI, reference interval; $r_p$, standardized partial regression coefficient; RSF, red blood cell size factor; RUO, research use only; RV, reference value; SD, standard deviation; SDR, : standard deviation ratio; Smk, smoking cigarettes; TIBC, total iron binding capacity; TRF, tran, sferrin; TRFSat, transferrin saturation; UIBC, unsaturated iron binding activity; UL–, upper limit; VitB12, Vitamin B12; WBC, white blood cell count.

countries, Russian RIs for total leukocyte and neutrophil counts were lower compared to most of other countries.

## Conclusion

RIs for the Russian population for 34 hematological and related parameters were established using up-to-date methods proposed by C-RIDL. Reducing the influences of latent anemia and obesity on RIs was crucial for erythrocyte parameters. Low levels of Russian RIs observed for platelet and neutrophil counts need further investigation.

## Introduction

Interpretation of laboratory test results starts with a comparison of values with reference intervals (RIs) adopted by the laboratory, which are usually those provided by manufacturers mostly derived for the American or European population and thus may not match local populations. CLSI EP 28 A3 guidelines recommend each laboratory to establish its own reference intervals [1]. With the recent advancement of standardization in laboratory tests, the consensus is to establish reference intervals for common use within a region or country by collaboration of multiple laboratories [2]. Accordingly, the IFCC Committee on Reference Intervals and Decision Limits (C-RIDL) developed an internationally harmonized protocol to facilitate country- or region-specific studies to define RIs [3]. It provides optimal techniques for recruitment, sampling, measurements, secondary exclusion, and statistical analyses to enable reproducible and standardized derivation of RIs. In 2012, C-RIDL launched a global multicenter study on RVs to promote the establishment of country-specific RIs. To make the derived RIs comparable across countries, a value-assigned serum panel was provided to the participating countries [4–13].

We collaborated with laboratories in three major cities in Russia to establish Russian population-specific RIs and explore sources of variations of RVs for the major chemistry analytes [12] and hormones and tumor markers measured by immunoassay [13]. This is our third report targeting RVs of hematological tests, including complete blood counts (CBC), iron markers, and related vitamins.

In the C-RIDL global project targeting biochemistry analytes, the latent abnormal values exclusion (LAVE) method was used to reduce the influence of inappropriate values, attributable to pre-sampling nutritional problems and/or muscular exertion. In the current study we investigated the utility of the LAVE method to reduce the influence of latent iron deficiency anemia in comparison to the conventional approach of explicit exclusion of individuals with low levels of serum ferritin and iron. Additionally, we compared Russian RIs with those reported from other countries.

## Materials and methods

### 1) The study protocol and recruitment

Healthy volunteers were recruited with the objectives of determining RIs for hematological and related parameters and exploring the source of variations of each analyte for better derivation and clinical interpretation of test results. The study protocol was prepared in harmonization to that provided by the IFCC, C-RIDL [3]. The inclusion and exclusion criteria adopted in the recruitment were as follows:

### Inclusion criteria

The participants were feeling well, between the ages of 18 and 65, and ideally, they were not taking any medication or supplements. However, if they were taking medication, doses and frequency were recorded accordingly.

### Exclusion criteria

The participants were excluded if any of the following were applicable: if he or she was diabetic and on oral therapy or insulin, had a history of chronic liver or kidney disease, had results from their blood samples that clearly point to a severe disease, had been a hospital in-patient or been subjectively seriously ill during the previous 4 weeks of participation, donated blood in the previous 3 months, if they were a known carrier of HBV, HCV, or HIV, if she was pregnant or within one year after childbirth and if they had participated in another research study involving an investigational product in the past 12 weeks.

This rather lenient criteria clearly entailed the need for secondary exclusion after measurements based on the pattern of the actual test results as described below. The health-status information acquired from the questionnaire (i.e., BMI, habits of regular exercise, drinking, or smoking, etc. [3]) was also used to consider for the secondary exclusion.

This study was conducted according to the Declaration of Helsinki and the study protocol was approved by the Ethics Committee of the City Hospital #40, Saint-Petersburg. The written informed consents were obtained from all participants.

The volunteers' recruitment period was started on December 19, 2013, and finished on December 18, 2014.

### 2) Sampling and measurements

A total of 506 healthy individuals >18 years of age were recruited in Saint-Petersburg (North-West region of Russia). The ratio of males to females was 46 to 54%. 99% of the recruited volunteers were Caucasians. Blood from each volunteer was collected into four 9 mL-vacuum tubes with clot activator and separation gel (Becton Dickinson, USA) and into one 5 mL-vacuum tube with K2-EDTA (Becton Dickinson, USA) for hematology analysis. Serum aliquots prepared from each volunteer were immediately frozen and then stored at −70˚C until the time of collective measurements.

### 3) Target analytes and measurements

In total, 34 hematology-related parameters were included in this study, which were grouped as follows:

**(3–1) Tests reflecting the metabolism of iron, vitamin B12 and folate:** iron (Fe), unsaturated iron-binding capacity (UIBC), total iron-binding capacity (TIBC), transferrin (TRF), transferrin saturation (TRFSat), ferritin (Ferr), vitamin B12 (VitB12), and folate;

**(3–2) Tests related to erythrocytes:** red blood cell count (RBC), hemoglobin (Hb), hematocrit (Ht), mean corpuscular volume (MCV), mean corpuscular hemoglobin (MCH), mean corpuscular hemoglobin concentration (MCHC), red blood cell distribution width in SD and in CV (RDW-SD, RDW-CV);

**(3–3) Tests related to immature erythrocytes:** reticulocyte count and percentage (Ret, Ret %), mean reticulocyte volume (MRV), immature reticulocyte fraction (IRF);

**(3–4) Tests related to leukocytes**: white blood cell count (WBC), neutrophil count and percentage (Neu, Neu%), eosinophil count and percentage (Eos, Eos%), basophil count and

percentage (Bas, Bas%), monocytes count and percentage (Mon, Mon%), lymphocytes count and percentage (Lym, Lym%);

**(3–5) Tests related to platelet**: platelet count (PLT), mean platelet volume (MPV), and platelet distribution width (PDW).

The measurements of Fe, UIBC, TIBC, TRF, and TRFSat were performed on the biochemistry analyzer AU 5800 (Beckman Coulter Inc., USA); ferritin, Vit B12, and folate were measured on the immunochemistry analyzer UniCel DxI 800 (Beckman Coulter Inc., USA) according to the requirements provided in the manufacturer's instructions for each assay. Hematological tests, or so-called CBC, were performed using the hematology analyzer UniCel DxH 800 (Beckman Coulter Inc., USA) on the day of blood collection. Description of assay method, calibrator and traceability is provided in **S1 Table in S1 File**.

## 4) Quality control

All laboratory measurements were done according to the laboratory's Quality Manual and standard operating procedures (SOPs). Instructions described in the manuals of each analyzer and in the reagent/kit package inserts were adhered to.

For clinical chemistry and immunochemistry tests, quality control was performed in two ways. One was through twice-daily measurement of 2 or 3 levels of QC specimens obtained from Beckman Coulter Inc. and Bio-Rad Laboratories, Inc., and the other was through daily measurement of a mini panel, composed of six sera from healthy volunteers (3 women and 3 men), as described in the common protocol. Based on repeated mini panel measurements, between-day coefficient of variation (CV) was calculated for each analyte and compared to desirable and minimal within-individual CV (CVi%), respectively corresponding to 1/2 and 3/4 of CVi%, published by the European Federation of Laboratory Medicine (EFLM) on the Biological Variation Database website (https://biologicalvariation.eu/). All the analytes met EFLM specifications for the desirable CV except Vitamin B12 (CV = 5.4%), which, however, met the minimum allowance level of 3/4 CVi = 5.4% (**S1 Table in S1 File**).

Additionally, as a part of the study objectives, a worldwide comparison of RVs was performed based on the test results of a serum panel consisting of 40 specimens provided by C-RIDL, measured in four batches over a period of four days.

For CBC parameters, special control materials COULTER 6C cell and Retic-X cell provided by the manufacturer were used for monitoring between-day analytical variations.

## 5) Statistical methods

Data analyses and statistical methods used were those recommended in the C-RIDL protocol [3, 14, 15]. In brief, the procedures are as follows:

**5.1) Sources of variation of reference values.** The multiple regression analysis (MRA) was used to assess the individual contribution of variation factors on each analyte. It was performed by setting the RVs of each analyte as an objective variable and the following factors as a fixed set of explanatory variables: sex, age, body mass index (BMI), the levels (see below) of cigarette smoking, alcohol consumption, and regular physical exercise, which were obtained from the questionnaire. The smoking, alcohol consumption, and regular exercise levels were classified into 2, 5, and 8 categories, respectively, using the following boundaries: 0 = no, 1 = yes; none, <12.5, 12.5–25, 25–50, >50 g ethanol/day; none, 1–7 days/week. The association of each factor on RVs was expressed using a standardized partial regression coefficient ($r_p$), which corresponds to the partial correlation coefficient, taking a value between −1.0 and 1.0. Since statistical testing of $r_p$ is too sensitive with large data size, we interpreted its practical significance ("effect size") in reference to the Cohen's guide [16] as small $0.1 \leq |r_p| < 0.3$; medium

$0.3 \leq |r_P| < 0.5$, large $|r_P| \geq 0.5$. Hence, we regarded $|0.3| \leq r_P$ as practically significant when considering the influence of a given factor on the RVs.

**5.2) Criteria for partitioning RVs.** The standard deviation ratio (SDR) was used as a criterion for assessing the need of partitioning RVs by sex and age. To calculate SDR, pure components of between-sex SD ($SD_{sex}$) and between-age SD ($SD_{age}$) were first computed by two-level nested ANOVA and within-group SD (approximately 1/4 the width of RI, or $SD_{RI}$). The SDRs for sex or age ($SDR_{sex}$, $SDR_{age}$) were calculated by taking respective ratios of $SD_{sex}$ and $SD_{age}$ to $SD_{RI}$. Since $SDR_{age}$ often differs between sexes, it was also calculated by one-way ANOVA after partitioning RVs by male and female, as $SDR_{ageM}$, $SDR_{ageF}$. The need for partition of RVs was considered by setting $SDR \geq 0.4$ as a primary guide: i.e., the threshold of 0.4 was inferred from conventional clinical practice of setting sex-specific RIs [14].

However, SDR can be too sensitive, when the width of RI constituting the denominator of SDR is narrow. Conversely, SDR may be insensitive, when between-subgroup differences occur mainly at the periphery of distribution (LL or UL). In general, SDR represents between-subgroup differences at the center of the distributions. Therefore, we additionally considered actual difference (bias) at LL or UL as "bias ratio" (BR) using the following formula, illustrating a case of assessing gender difference:

$$\mathrm{BR_{LL}} = \frac{|LL_M - LL_F|}{(UL_{MF} - LL_{MF})/3.92}, \ \mathrm{BR_{UL}} = \frac{|UL_M - UL_F|}{(UL_{MF} - LL_{MF})/3.92}$$

where subscript M, F, and MF represent male, female, and male+female, respectively. The denominator of each formula represents the standard deviation ($SD_{RI}$) comprising the RI, the width of which corresponds to 3.92 times $SD_{RI}$. To make the threshold of BR aligned to SDR with the common denominator ($SD_{RI}$), we multiplied the threshold of SDR = 0.4 by $\sqrt{2}$ to obtain BR's threshold of 0.57 (i.e., absolute difference (bias) of two values ($x_1$ and $x_2$), or $|x_1 - x_2|$, is $\sqrt{2}$ times larger than SD of the two values).

Whereas, when BR was used for judging the effectiveness of the LAVE method, its threshold was set according to the conventional specification of allowable analytical bias (= difference) of a minimum level: $0.375 \times \sqrt{SD_G^2 + SD_I^2}$ ($= SD_{RI}$) [13]. Hence 0.375 was set as the threshold for BR in contrast to the above-mentioned threshold of BR for between-sex bias, which was inferred from conventional clinical practice.

**5.3) Derivation of reference intervals.** RIs were derived by both parametric and nonparametric methods for comparative purposes. The former was performed after the Gaussian transformation of RVs using the two-parameter Box-Cox power transformation formula [17]. The 90% confidence interval (CI) of the lower limit (LL) and upper limit (UL) of RI was calculated by the bootstrap method through random resampling of the same dataset 50 times. Accordingly, the final LL and UL of RI, derived both by parametric and nonparametric methods, were chosen as the average of iteratively derived LLs and ULs. The RIs derived by the nonparametric method were adopted only when the parametric method failed in Gaussian transformation.

To reduce the influence of latent iron deficiency, the following two methods of secondary exclusion were implemented and compared:

1) Latent abnormal values exclusion (LAVE) method with an iterative exclusion of individuals with abnormal results among reference tests associated with erythrocyte parameters (**S1 Fig**). Two sets of reference tests for the LAVE method were chosen based on Spearman's correlation matrix shown in **S2 Table in S1 File** [see Discussion] for comparative purposes:

#1: Seven reference tests with iron markers: Hb, MCH, MCHC, RDW-CV, Ret, Fe, and ferritin.

#2: Five reference tests without iron markers: Hb, MCH, MCHC, RDW-CV, and Ret

In both schemes, no abnormal results among the reference tests were allowed, but the RIs of respective tests were empirically extended to 3% of the width of the RI on both limits (i.e., LL' = LL −0.03× (UL−LL)/3.92; UL' = UL +0.03× (UL−LL)/3.92) so as not to delete too many records. While the number of iterative derivations was set to 6.

2)) Explicit exclusion of individuals with IDA as defined, with either ferritin or Fe being below respective decision limit, provided in the Access Ferritin reagent instructions for use (IFU), and for the AU Fe reagent (Beckman Coulter Inc.) (i.e., for ferritin <23.9 μg/L (male) and <11 μg/L (female); for Fe<12.5 μmol/L (male) and <10.7 μmol/L (female)) [18, 19].

For the leukocyte and platelet-related parameters, we performed a preliminary analysis on the effectiveness of the LAVE method in reducing the influence of latent infectious conditions on their test results in reference to the results of inflammatory markers: TF, and ferritin. However, the LAVE method did not cause much change in reference limits. Spearman's correlation coefficient matrix built with leukocyte and platelet parameters did not show many cross-associations among them. These findings led us to interpret that subclinical infectious conditions among healthy subjects were not as frequent as to affect the derivation of RIs for those analytes. Hence, we did not apply the LAVE method to the analytes related to leukocytes and platelets (**S2 Table in S1 File**).

## Results

### 1. Sources of variation of reference values

Sources of variations of each analyte were evaluated by MRA. The magnitude of associations of 5-factors (age, BMI, levels of drinking, smoking, and regular exercise) with each of 34 parameters were computed by MRA as $r_p$ and listed in **Table 1**. By setting $|r_p| \geq 0.3$ as practical importance, positive association of RVs with age was observed for 2 analytes in males, including MCV and RDW-SD ($r_p = 0.306$ and $0.376$, respectively) and for 4 analytes in females, with age-related decrease of RVs for TIBC, TF, and Mon (−0.339, −0.330, and −0.320, respectively) and age-related increase of RVs for ferritin (0.391). A significant BMI-related increase in RVs was observed for Ret, Ret% and RBC in both sexes ($r_p = 0.467$, $0.431$, and $0.302$ for males; 0.591, 0.543, and 0.308 for females). In males, BMI-related increase was also observed for TIBC and TF (0.331 and 0.313), in females for IRF (0.429). In contrast, no apparent association of alcohol consumption, smoking and physical activities was found based on the MRA.

### 2. Criteria for partitioning RVs

$SDR_{sex}$, a magnitude of between-sex differences in RVs, was computed by 2-level nested ANOVA in two ways, either with age or BMI as a covariate. The between-age and between-BMI differences in RVs were computed as $SDR_{age}$ and $SDR_{BMI}$ by one-way ANOVA, separately for each sex (**Tables 2 and 3**), where age and BMI were subgrouped respectively at the boundary of 30, 40, and 50 years of age, and 20, 25, and 30 kg/m$^2$. By setting $SDR \geq 0.4$ as a threshold, the sex-related differences were significant in eight analytes: UIBC, TF, TFRSat, ferritin, RBC, Hb, Ht, and MCHC ($SDR_{sex} = 0.478$, $0.426$, $0.429$, $1.252$, $1.106$, $1.381$, $1.369$, and $0.460$ respectively).

Between-gender differences are clearly observed for 20 analytes as shown in **Fig 1**. For age-related changes, significant differences were observed for the following analytes: ferritin in females ($SDR_{age} = 0.656$), and RDW-SD in males ($SDR_{age} = 0.47$), as illustrated in **Fig 2**. BMI-related differences in RVs were significant for Ret, Ret% and IRF: i.e., $SDR_{BMI} = 0.691$, $0.625$, and $0.451$ respectively, calculated for both genders with 2-level nested ANOVA. While $SDR_{BMI}$ calculated for each gender by one-way ANOVA for Ret and Ret% were 0.602 and 0.568, respectively, in males, and 0.743 and 0.657, respectively, in females; for IRF and RBC

**Table 1. Results of multiple regression analysis for sources of variation of RVs.**

| Analyte | Male | | | | | | | Female | | | | | | |
|---|---|---|---|---|---|---|---|---|---|---|---|---|---|---|
| | n | R | Age | BMI | DrkLvl | SmkLvl | ExerLvl | n | R | Age | BMI | DrkLvl | SmkLvl | ExerLvl |
| Fe | 224 | 0.138 | 0.110 | 0.001 | -0.097 | 0.038 | -0.026 | 262 | 0.211 | 0.038 | -0.031 | 0.192 | 0.057 | -0.029 |
| UIBC | 193 | 0.245 | -0.066 | 0.247 | 0.030 | -0.012 | -0.042 | 225 | 0.279 | -0.241 | 0.128 | -0.169 | 0.026 | 0.093 |
| TIBC | 191 | 0.334 | -0.087 | **0.331** | -0.069 | 0.009 | -0.005 | 225 | 0.327 | **-0.339** | 0.189 | -0.098 | 0.059 | 0.093 |
| TF | 221 | 0.308 | -0.110 | **0.313** | -0.036 | 0.020 | -0.024 | 261 | 0.331 | **-0.330** | 0.199 | -0.115 | 0.098 | 0.053 |
| TRFSat | 192 | 0.123 | 0.089 | -0.077 | -0.056 | 0.046 | 0.022 | 223 | 0.228 | 0.133 | -0.035 | 0.188 | 0.018 | -0.060 |
| Ferr | 221 | 0.239 | -0.086 | 0.173 | 0.129 | -0.079 | -0.088 | 262 | 0.580 | **0.391** | **0.252** | 0.176 | 0.066 | -0.068 |
| VitB12 | 221 | 0.253 | 0.031 | 0.123 | -0.061 | -0.013 | 0.194 | 257 | 0.215 | 0.098 | 0.092 | 0.010 | -0.005 | 0.140 |
| Folate | 223 | 0.313 | 0.198 | 0.104 | -0.001 | -0.117 | 0.117 | 261 | 0.129 | 0.006 | 0.073 | 0.011 | -0.010 | 0.100 |
| RBC | 223 | 0.421 | -0.155 | **0.302** | -0.219 | -0.084 | -0.201 | 262 | 0.298 | -0.040 | **0.308** | -0.045 | -0.016 | 0.016 |
| Hb | 224 | 0.339 | -0.010 | **0.265** | -0.182 | 0.077 | -0.144 | 261 | 0.341 | 0.166 | 0.225 | 0.057 | 0.056 | -0.005 |
| Ht | 224 | 0.331 | 0.022 | **0.260** | -0.158 | 0.097 | -0.138 | 259 | 0.336 | 0.116 | **0.263** | 0.039 | 0.054 | 0.008 |
| MCV | 223 | 0.416 | **0.306** | -0.181 | 0.064 | 0.235 | 0.117 | 251 | 0.302 | 0.204 | 0.017 | 0.124 | 0.172 | 0.077 |
| MCH | 220 | 0.345 | **0.254** | -0.169 | 0.016 | 0.183 | 0.137 | 243 | 0.266 | **0.256** | -0.034 | 0.086 | 0.097 | 0.045 |
| MCHC | 219 | 0.223 | -0.137 | -0.003 | -0.082 | -0.133 | 0.009 | 257 | 0.217 | 0.233 | -0.055 | 0.038 | 0.006 | -0.035 |
| RDW-SD | 222 | 0.484 | **0.376** | -0.085 | 0.114 | **0.261** | 0.065 | 260 | 0.138 | 0.098 | 0.018 | -0.009 | 0.097 | -0.013 |
| RDW-CV | 224 | 0.308 | 0.218 | 0.089 | 0.035 | 0.158 | -0.010 | 246 | 0.146 | -0.079 | -0.029 | -0.057 | -0.074 | -0.064 |
| Ret | 162 | 0.464 | -0.089 | **0.467** | -0.036 | -0.036 | -0.115 | 233 | 0.585 | -0.017 | **0.591** | 0.014 | 0.075 | 0.026 |
| Ret% | 161 | 0.424 | -0.057 | **0.431** | 0.012 | -0.025 | -0.083 | 232 | 0.542 | -0.008 | **0.543** | 0.033 | 0.073 | 0.031 |
| MRV | 160 | 0.302 | **0.255** | -0.178 | 0.043 | 0.120 | 0.065 | 233 | 0.230 | 0.161 | -0.171 | 0.084 | 0.107 | 0.094 |
| IRF | 163 | 0.320 | 0.090 | 0.220 | 0.008 | 0.151 | 0.123 | 233 | 0.432 | -0.003 | **0.429** | 0.054 | 0.078 | -0.021 |
| WBC | 199 | 0.384 | -0.011 | 0.231 | -0.167 | 0.217 | -0.166 | 259 | 0.278 | -0.233 | **0.275** | -0.007 | 0.051 | -0.099 |
| Neu | 220 | 0.323 | 0.060 | 0.197 | -0.091 | 0.203 | -0.113 | 259 | 0.259 | -0.168 | **0.269** | 0.018 | 0.081 | -0.075 |
| Eos | 201 | 0.225 | -0.063 | 0.026 | -0.057 | 0.173 | -0.081 | 260 | 0.155 | -0.060 | 0.121 | -0.007 | -0.058 | -0.103 |
| Bas | 201 | 0.167 | 0.098 | 0.023 | -0.122 | 0.089 | -0.019 | 262 | 0.178 | -0.192 | 0.060 | -0.023 | -0.058 | -0.033 |
| Mon | 199 | 0.274 | -0.015 | 0.135 | -0.161 | 0.193 | -0.028 | 262 | 0.342 | **-0.320** | **0.287** | -0.002 | 0.006 | -0.163 |
| Lym | 199 | 0.325 | -0.155 | 0.202 | -0.131 | 0.069 | -0.178 | 261 | 0.279 | **-0.279** | 0.203 | -0.069 | 0.041 | -0.096 |
| Neu% | 223 | 0.224 | 0.170 | 0.045 | -0.007 | 0.129 | 0.025 | 261 | 0.184 | -0.013 | 0.150 | 0.046 | 0.105 | 0.015 |
| Eos% | 224 | 0.140 | -0.056 | -0.068 | 0.028 | 0.080 | -0.039 | 260 | 0.123 | 0.049 | -0.010 | 0.005 | -0.100 | -0.049 |
| Bas% | 221 | 0.128 | 0.065 | -0.025 | -0.052 | -0.101 | 0.014 | 259 | 0.181 | -0.044 | -0.129 | -0.028 | -0.096 | 0.023 |
| Mon% | 221 | 0.146 | 0.028 | -0.025 | -0.051 | 0.007 | 0.134 | 261 | 0.131 | -0.095 | 0.020 | 0.011 | -0.093 | -0.064 |
| Lym% | 224 | 0.268 | -0.200 | -0.001 | 0.033 | -0.182 | -0.058 | 261 | 0.178 | -0.039 | -0.125 | -0.080 | -0.063 | 0.015 |
| PLT | 224 | 0.135 | -0.036 | -0.005 | -0.003 | -0.014 | -0.130 | 262 | 0.225 | -0.208 | 0.188 | 0.042 | -0.014 | -0.110 |
| MPV | 223 | 0.289 | **0.258** | -0.060 | -0.071 | 0.078 | -0.110 | 262 | 0.133 | -0.093 | 0.100 | 0.006 | 0.049 | 0.068 |
| PDW | 222 | 0.251 | -0.181 | 0.187 | 0.011 | 0.010 | 0.101 | 262 | 0.064 | 0.038 | -0.021 | -0.022 | 0.004 | -0.046 |

The listed values are standardized partial correlation coefficients ($r_P$). $|r_P| > 0.25$ were shown in bold, and $|r_P| > 0.30$ were marked in orange (+) or blue (-) shade in two grades, as exceeding the effect size.

R: multiple regression coefficient; BMI: body-mass index; DrkLvl, SmkLvl, ExerLvl: levels of drinking, smoking, and regular exercise habits.

$SDR_{BMI}$ = 0.523 and 0.451, respectively, in females and males. BMI-related changes in RVs that were significant according to MRA ($r_P \geq 0.3$) are illustrated for RBC, Ret, Ret%, and IRF in **Fig 2**, in which BMI values were stratified at $< 20$, 20–25, 25–30, $\geq 30$ kg/m$^2$.

## 3. Derivation of reference intervals

To find an optimal method for secondary exclusion of latent anemia, the following four methods were compared for derivation of RIs of erythrocyte parameters:

**Table 2. SDR for the magnitude of sex and age -related changes in hematology RVs.**

| | 2N-ANOVA | | 1way-ANOVA | | | 2N-ANOVA | | 1way-ANOVA | |
|---|---|---|---|---|---|---|---|---|---|
| Analyte | SDRsex | SDRage | SDRageM | SDRageF | Analyte | SDRsex | SDRage | SDRageM | SDRageF |
| Fe* | **0.356** | 0.000 | 0.000 | 0.000 | Ret%* | 0.000 | 0.223 | 0.030 | 0.284 |
| UIBC* | 0.478 | 0.045 | 0.000 | 0.124 | MRV | 0.000 | 0.161 | 0.234 | 0.100 |
| TIBC* | **0.338** | 0.206 | 0.000 | 0.265 | IRF | 0.000 | 0.270 | 0.255 | 0.279 |
| TF* | 0.426 | 0.205 | 0.000 | 0.272 | WBC* | 0.167 | 0.000 | 0.000 | 0.042 |
| TRFSat* | 0.429 | 0.000 | 0.000 | 0.000 | Neu* | 0.000 | 0.001 | 0.000 | 0.037 |
| Ferr* | 1.252 | 0.491 | 0.000 | 0.656 | Eos* | 0.219 | 0.118 | 0.043 | 0.158 |
| VitB12* | 0.175 | 0.172 | 0.186 | 0.161 | Bas* | 0.000 | 0.149 | 0.023 | 0.203 |
| Folate* | 0.276 | 0.182 | 0.213 | 0.152 | Mon* | 0.414 | 0.116 | 0.000 | 0.201 |
| RBC | 1.106 | 0.162 | 0.177 | 0.148 | Lym* | 0.262 | 0.171 | 0.112 | 0.205 |
| Hb* | 1.381 | 0.237 | 0.000 | **0.334** | Neu% | 0.250 | 0.169 | 0.193 | 0.149 |
| Ht | 1.369 | 0.223 | 0.071 | 0.280 | Eos%* | 0.172 | 0.102 | 0.068 | 0.130 |
| MCV | 0.061 | 0.256 | **0.308** | 0.216 | Bas% | 0.134 | 0.031 | 0.000 | 0.109 |
| MCH | 0.212 | 0.235 | 0.224 | 0.243 | Mon% | **0.367** | 0.000 | 0.000 | 0.000 |
| MCHC | 0.460 | 0.249 | 0.264 | 0.241 | Lym% | 0.099 | 0.205 | 0.228 | 0.185 |
| RDW-SD* | 0.000 | **0.337** | 0.470 | 0.145 | PLT | **0.317** | 0.046 | 0.060 | 0.037 |
| RDW-CV* | 0.256 | 0.143 | 0.292 | 0.000 | MPV* | 0.129 | 0.237 | **0.327** | 0.063 |
| Ret* | **0.339** | 0.226 | 0.000 | **0.305** | PDW | 0.171 | 0.158 | 0.147 | 0.164 |

2N-ANOVA: two-level nested ANOVA; SDRsex, SDRage: SD ratio for between-sex and between-age variations

SDR values that exceeded 0.3 were shown in bold font, and those exceeded 0.4 (the threshold) or 0.6 were highlighted by light orange and orange shades, respectively.

* symbol indicates calculation of SDR after logarithmic transformation of reference values.

1. No exclusion was performed;

2. LAVE method using 7 reference tests (Fe, Ferr, Hb, MCH, MCHC, RDW-CV, Ret);

3. LAVE method using 5 reference tests (Hb, MCH, MCHC, RDW-CV, Ret);

4. Explicit exclusion of patients who belonged to IDA group by the criteria described in the Methods.

The distribution of RV datasets after selection by 1) to 4) are shown in **Fig 3**. for six representative analytes as a scattergram located in the 1st, 2nd, 4th, and 6th row, respectively.

**Table 3. SDR for the magnitude of sex and BMI -related changes in hematology RVs.**

| | 2N-ANOVA | | 1way-ANOVA | |
|---|---|---|---|---|
| Analyte | SDRsex | SDR$_{BMI}$ | SDR$_{BMI}$M | SDR$_{BMI}$F |
| RBC | **1.134** | **0.375** | **0.340** | 0.402 |
| Hb* | **1.398** | **0.331** | 0.249 | **0.376** |
| Ret* | 0.168 | **0.691** | **0.602** | **0.743** |
| Ret%* | 0.000 | **0.625** | **0.568** | **0.657** |
| IRF | 0.000 | **0.451** | **0.310** | **0.523** |

2N-ANOVA: two-level nested ANOVA; SDRsex, SDRBMI: SD ratio for between-sex and between-BMI variations. BMI was partitioned at 20, 25, 30 kg/m² in the calculation.

SDR values that exceeded 0.3 were shown in bold font, and those exceeded 0.4 (the threshold) or 0.6 were highlighted by light orange and orange shades, respectively.

* symbol indicates calculation of SDR after logarithmic transformation of reference values.

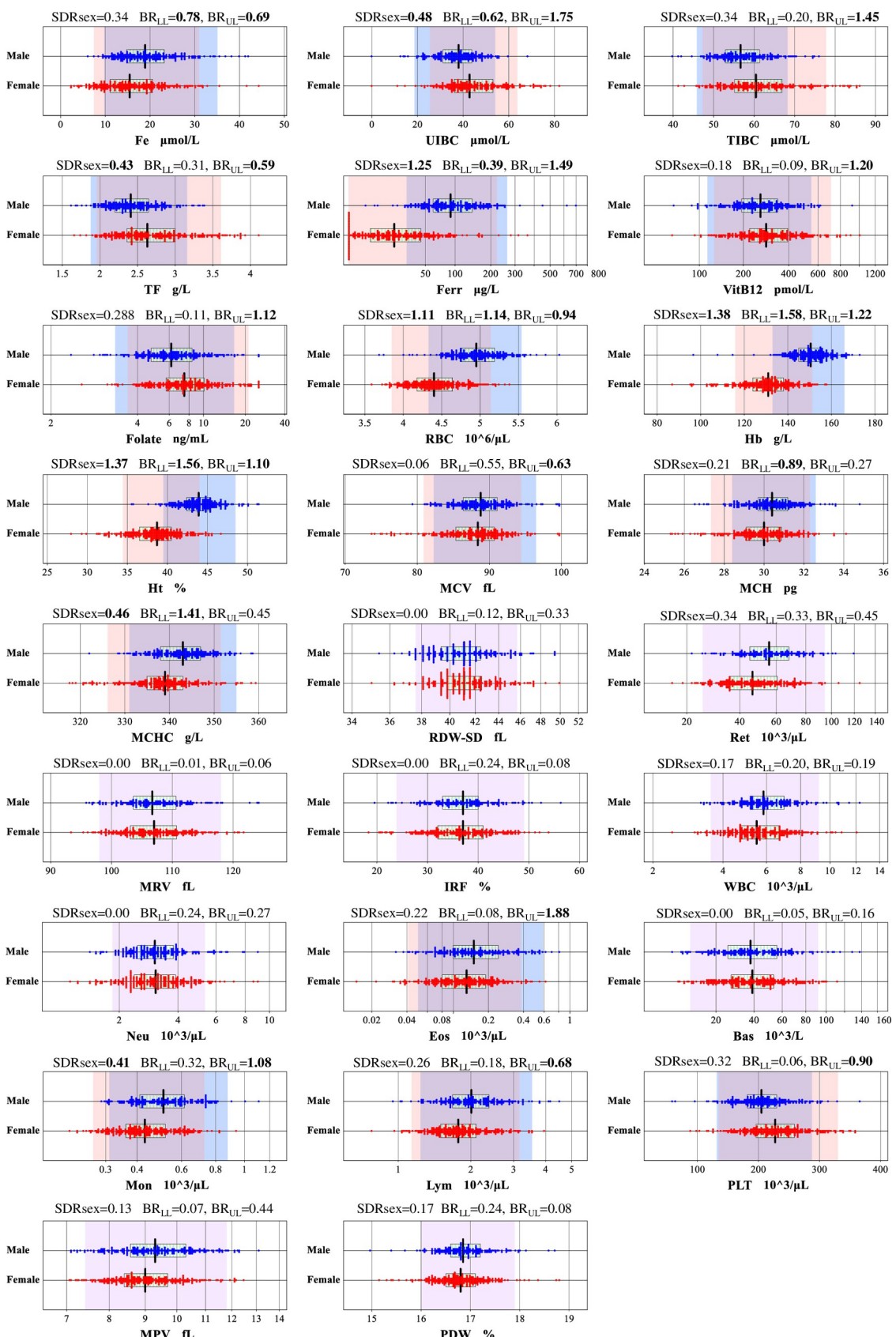

**Fig 1. Between-gender differences for major parameters.** Gender-specific distributions of RVs for major parameters are drawn by scattergrams. The background shades represent RIs determined for male (blue), female (pink), and for both (purple). The central 50% range and median of RVs are shown by the horizontal box and the vertical line.

As a reference, the following three datasets were prepared and plotted in the 3rd, 5th, and 7th rows: i.e., 3rd for datasets of patients with abnormal results among 7 reference tests, 5th for patients with abnormal results among 5 reference tests, and 7th for patients belonging to the IDA group.

As a result, the fifth set of reference tests (Hb, MCH, MCHC, RDW-CV, and Ret) was found most efficient in terms of adjusting the effect of RI limits for erythrocyte parameters, iron markers and related vitamins, and the data size that remained after applying the LAVE method (i.e., the use of seven reference tests caused more reduction in sample size for calculating RIs, despite minor differences in RI limits), while the exclusion based on IDA criteria was regarded less efficient in adjusting RI limits.

The effect of partitioning RVs by sex and age on RIs is summarized in **S3 Table in S1 File** for erythrocyte parameters, iron markers and related vitamins, and in **S4 Table in S1 File** for

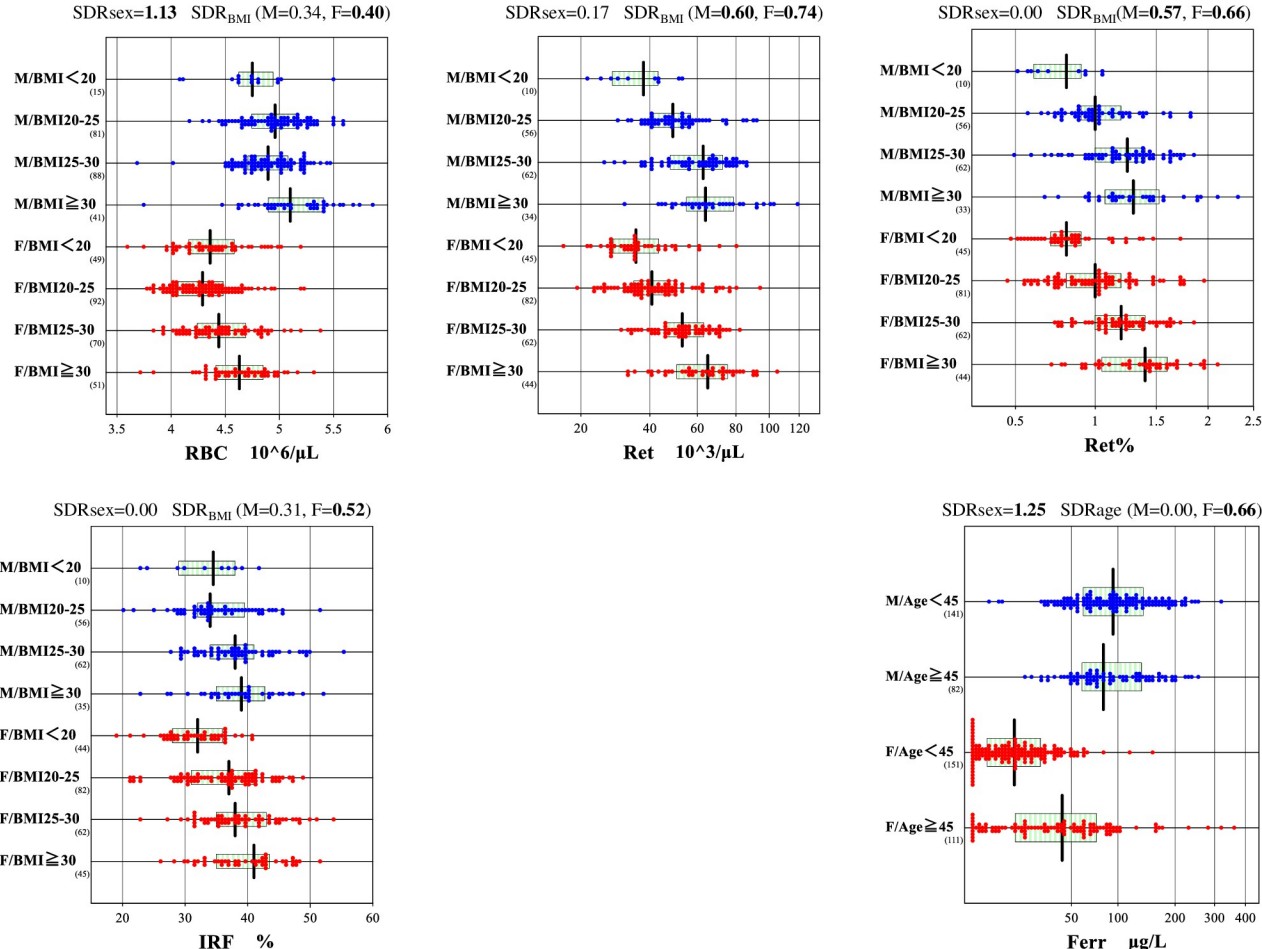

**Fig 2. Hematology RVs influenced by BMI and age.** RVs of RBC, Ret, Ret%, and IFR were partitioned by gender and BMI into five groups, while RVs were partitioned by age at 45 years. The central 50% range and median of RVs are shown by the horizontal box and the vertical line. No secondary exclusion was done while drawing the scattergrams.

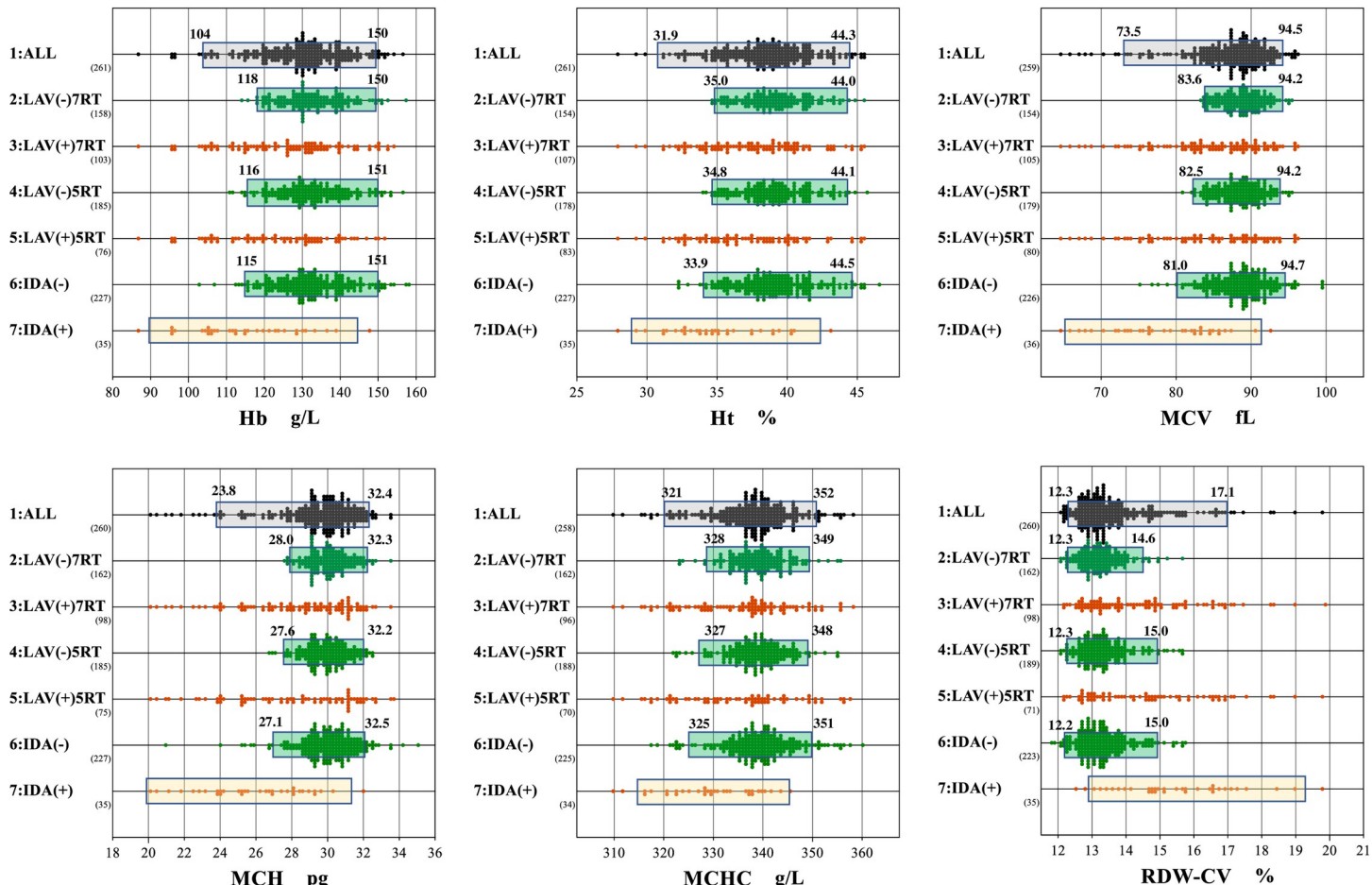

**Fig 3. The effect of LAVE on RIs for hematology tests in comparison to IDA criteria.** Female RIs of 6 representative hematology tests were determined parametrically using the following 4 sets of RV datasets. 1: All: RVs without any prior exclusion. 2: LAVE(-) 7RT: RVs with no abnormal value among 7 tests (Hb, MCH, MCHC, RDW-CV, Ret, Fe, Ferr) other than itself. 4: LAVE(-) 5RT: RVs with no abnormal value among 5 ref tests (Hb, MCH, MCHC, RDW-CV, Ret) other than itself. 6: IDA(-): RVs with no abnormal results for Fe and Ferr from respective lower limit of the manufacturer's RI. Each bar shown over the scattergram represents the span of RI with its lower and upper limits indicated by numerical labels. Please note that RIs indicated by green shade (at the 4th row) correspond to what we adopted in this study. The three scattergrams of RVs shown in red dots in the 3rd, 5th, and 7th rows represent values that were excluded by the criteria applied to obtain RVs shown above in the 2nd, 4th, and 6th rows in green. Thus, their group labels were designated as follows: As a reference, the following three datasets were prepared and plotted in the 3rd, 5th, and 7th row. 3: LAVE(+) 7RT: RVs with one or more abnormal values among 7 ref tests (Hb, MCH, MCHC, RDW-CV, Ret, Fe, Ferr). 5: LAVE(+) 5RT: RVs with one or more abnormal values among 5 ref tests (Hb, MCH, MCHC, RDW-CV, Ret). 7: IDA(+): RVs had abnormal results either for Fe or Ferr.

leukocytes and platelets. The choice of partitioning by sex and age was primary decided based on SDR>0.4, $r_p$>0.3, and the effect of partition was also assessed from $BR_{LL}$ or $BR_{UL,}$ by setting its threshold of >0.57. Based on this information, partitioning RVs by sex was performed for 19 analytes, including RBC, Hb, Ht, MCV, RDW-CV, Fe, UIBC, TIBC, TF, TFRSat, ferritin, VitB12, Folate, Eos, Eos%, Mon, Mon%, Lym, and PLT. RIs were partitioned by age for ferritin in females (**S5 Table in S1 File**).

Among analytes with BMI-related changes (Hb, Ht in females, and RBC, Ret, Ret% in both sexes), exclusion of individuals with BMI>28 was found to be effective in lowering UL judged from $BR_{UL}$ (**S6 Table in S1 File**).

Regarding the effect of the LAVE method on RI limits (by setting five reference tests: Hb, MCH, MCHC, RDW-SD, and Ret), which is shown in **S3 Table in S1 File**, we regarded the following tests as "LAVE-sensitive": Hb, MCV, RDW-CV, Fe, UIBC, TF (<45 years) in females, RBC, Ht, ferritin in males, and MCH, MCHC, RDW-SD, TIBC, TRFSat in both sexes.

After applying all these schemes of partitioning RVs by sex or age, and of secondary exclusion of RVs by LAVE and/or BMI restriction, the final RIs for all the 34 parameters were listed in **Table 4**.

## Discussion

The main goals of the current study were the following: 1) to explore sources of variation of RVs (sex, age, and BMI) for better interpretation of hematology test results, 2) to derive RIs for a heterogeneous group of tests in the most rational way by applying an analyte-specific scheme of partitioning, and the secondary exclusion of latent diseases, and 3) to grasp the international perspective of Russian RIs by comparing to those reported in other studies or provided in clinical guidelines.

### 1. BMI-related changes in RVs

RVs of ferritin were strongly associated with age in females (SDRage = 0.656). This age-related increase in ferritin is well-known and can be explained by the cessation of menstrual losses. Ferritin is an inflammatory marker, and its level increases in elderly females [20].

A significant BMI-related increase in RVs of Ret and Ret% was conspicuously observed in both sexes ($SDR_{BMI}$ = 0.60 and 0.57 in males; 0.74 and 0.66 in females, respectively). This finding wasn't considered to be confounded by the age-related increase in BMI, because MRA showed an independent association of BMI with Ret and Ret% ($r_p$ = 0.47 and 0.43 in males; 0.59 and 0.54 in females, respectively) after adjusting for age. The effect of BMI was also detected in a lesser degree for RBC in both sexes, and for Hb and Ht in females. It could be explained by recent findings regarding the role of hepcidin, a hormonal peptide secreted by the liver and adipocytes, in regulating iron homeostasis and erythrocyte production. High hepcidin levels block intestinal iron absorption and macrophage iron recycling, causing iron-restricted erythropoiesis and anemia [21]. These observations would be relevant in patients with increased BMI and a large volume of adipose tissue. Perez et al. have also reported that the hemoglobin and erythrocyte indices did not vary by body weight alone but suggested that obesity with its resultant inflammatory state can lead to iron deficiency, which, in turn, can induce reticulocytosis [22]. In this regard, Jeong et al., have demonstrated a positive association of RBC count, Hb, and Ht levels with BMI [23]. This observation is compatible with a report by He et al., that Hb could be useful in predicting a new onset of metabolic syndrome [24].

On the other hand, several studies had reported that Hb levels were significantly associated with high blood pressure. Shimizu et al., showed that Ret levels in elderly Japanese individuals were positively associated with hypertension, which is known to be associated with BMI. They postulated that the increased BMI caused oxidative stress in arterial walls, and might induce hematopoiesis and reticulocytosis, which was regarded as having an antioxidant effect [25].

Finally, Christakoudi et al. published a large cross-sectional retrospective study, where 105,853 women and 100,854 men from UK biobank cohort were enrolled. The authors showed clear positive association between BMI and all parameters, including Hb, Ht, and erythrocyte count, and more strongly with total reticulocyte count and percent, immature reticulocyte count and IRF, with little differences between women and men, or between pre-menopausal and postmenopausal women. These observations are quite consistent with our findings. Besides the effect of hepcidin, the investigators suggested the role of leptin in stimulating of erythropoiesis in obese patients, which was also produced by adipose cells. The authors suggested leptin as alternative parameter to a key factor stimulating erythropoiesis—erythropoietin (EPO). Thus, in chronic kidney disease, when EPO production is restricted due to kidney

**Table 4.** The list of RIs adopted for hematological parameters by sex and age.

| Item | Unit | LAVE | BMI | Sex | Age | N | LL_L | LL_H | LL | Me | UL | UL_L | UL_H |
|---|---|---|---|---|---|---|---|---|---|---|---|---|---|
| RBC | 10^6 /μL | LAVE(+) | <28 | M | All | 124 | 4.41 | 4.53 | 4.46 | 4.92 | 5.46 | 5.39 | 5.51 |
| | | LAVE(+) | <28 | F | All | 132 | 3.84 | 3.94 | 3.88 | 4.34 | 4.94 | 4.83 | 5.05 |
| Hb | g/L | LAVE(-) | All | M | All | 223 | 132 | 137 | 134 | 151 | 166 | 165 | 168 |
| | | LAVE(+) | <28 | F | All | 139 | 112 | 116 | 114 | 130 | 148 | 145 | 151 |
| Ht | % | LAVE(+) | All | M | All | 179 | 39.7 | 40.5 | 40.1 | 44.0 | 48.3 | 47.8 | 48.8 |
| | | LAVE(+) | <28 | F | All | 132 | 33.6 | 34.7 | 34.1 | 38.4 | 42.5 | 42.0 | 42.9 |
| MCV | fL | LAVE(-) | All | M | All | 224 | 81.4 | 82.8 | 82.3 | 88.6 | 96.5 | 95.6 | 98.4 |
| | | LAVE(+) | All | F | All | 179 | 81.9 | 83.2 | 82.5 | 88.5 | 94.2 | 93.7 | 94.8 |
| MCH | pg | LAVE(+) | All | MF | All | 364 | 27.8 | 28.2 | 28.2 | 30.3 | 32.7 | 32.5 | 32.9 |
| MCHC | g/L | LAVE(+) | All | MF | All | 370 | 327 | 331 | 330 | 341 | 353 | 352 | 355 |
| RDW-SD | fL | LAVE(+) | All | MF | All | 351 | 37.5 | 37.9 | 37.7 | 40.9 | 45.5 | 45.1 | 46.0 |
| RDW-CV | % | LAVE(-) | All | M | All | 222 | 12.0 | 12.2 | 12.2 | 13.1 | 14.2 | 14.2 | 14.5 |
| | | LAVE(+) | All | F | All | 189 | 12.2 | 12.4 | 12.3 | 13.2 | 15.0 | 14.9 | 15.5 |
| Ret | 10^9/L | LAVE(-) | <28 | MF | All | 270 | 21.3 | 24.8 | 23.0 | 45.6 | 84.1 | 79.8 | 87.1 |
| Ret% | % | LAVE(-) | <28 | MF | All | 269 | 0.52 | 0.57 | 0.54 | 1.01 | 1.77 | 1.70 | 1.85 |
| MRV | fL | LAVE(-) | All | MF | All | 392 | 96.9 | 98.2 | 97.6 | 106.8 | 117.8 | 117.1 | 119.3 |
| IRF | % | LAVE(-) | All | MF | All | 396 | 23.4 | 25.5 | 24.5 | 36.8 | 49.8 | 48.7 | 50.9 |
| Fe | μmol/L | LAVE(-) | All | M | All | 225 | 8.5 | 10.6 | 9.8 | 18.8 | 35.1 | 32.5 | 37.3 |
| | | LAVE(+) | All | F | All | 177 | 6.0 | 7.8 | 7.2 | 16.4 | 30.5 | 29.0 | 33.7 |
| UIBC | μmol/L | LAVE(-) | All | M | All | 195 | 17.4 | 21.6 | 18.7 | 37.8 | 54.1 | 52.7 | 57.5 |
| | | LAVE(+) | All | F | All | 155 | 22.8 | 27.9 | 26.2 | 42.9 | 63.5 | 60.2 | 65.8 |
| TIBC | μmol/L | LAVE(+) | All | M | All | 157 | 44.2 | 46.8 | 46.0 | 57.3 | 71.4 | 69.2 | 73.6 |
| | | LAVE(+) | All | F | All | 155 | 44.9 | 47.8 | 47.0 | 59.6 | 78.4 | 75.5 | 81.0 |
| TF | g/L | LAVE(-) | All | M | All | 225 | 1.80 | 1.92 | 1.88 | 2.42 | 3.16 | 3.07 | 3.22 |
| | | LAVE(+) | All | F | All | 178 | 1.85 | 1.98 | 1.94 | 2.59 | 3.57 | 3.43 | 3.70 |
| TRFSat | % | LAVE(+) | All | M | All | 157 | 12.9 | 18.1 | 16.6 | 33.1 | 64.3 | 58.7 | 67.6 |
| | | LAVE(+) | All | F | All | 155 | 10.2 | 13.3 | 11.5 | 27.5 | 51.3 | 48.4 | 56.7 |
| Ferr | μg/L | LAVE(+) | All | M | All | 173 | 20.9 | 32.6 | 29.8 | 97 | 271 | 251 | 381 |
| | | LAVE(-) | All | F | ~45 | 150 | 4.7 | 5.8 | 5.5 | 15 | 67 | 56 | 94 |
| | | LAVE(-) | All | F | 45~ | 111 | 4.7 | 5.9 | 5.3 | 37 | 223 | 174 | 261 |
| VitB12 | pmol/L | LAVE(-) | All | M | All | 223 | 99 | 127 | 113 | 263 | 549 | 519 | 653 |
| | | LAVE(-) | All | F | All | 259 | 110 | 132 | 125 | 293 | 710 | 647 | 761 |
| Folate | ng/mL | LAVE(-) | All | M | All | 225 | 2.78 | 3.31 | 3.15 | 6.22 | 16.5 | 14.1 | 18.0 |
| | | LAVE(-) | All | F | All | 261 | 3.29 | 3.69 | 3.58 | 7.51 | 21.1 | 18.3 | 22.7 |
| WBC | 10^3 /μL | LAVE(-) | All | MF | All | 457 | 3.37 | 3.63 | 3.54 | 5.71 | 9.21 | 8.86 | 9.72 |
| Neu | 10^3 /μL | LAVE(-) | All | MF | All | 480 | 1.53 | 1.70 | 1.67 | 3.06 | 5.85 | 5.73 | 6.42 |
| Eos | 10^3 /μL | LAVE(-) | All | M | All | 200 | 0.042 | 0.054 | 0.050 | 0.15 | 0.59 | 0.524 | 0.689 |
| | | LAVE(-) | All | F | All | 257 | 0.033 | 0.043 | 0.041 | 0.13 | 0.38 | 0.342 | 0.430 |
| Bas | 10^3 /μL | LAVE(-) | All | MF | All | 460 | 0.010 | 0.012 | 0.011 | 0.039 | 0.092 | 0.090 | 0.100 |
| Mon | 10^3 /μL | LAVE(-) | All | M | All | 200 | 0.30 | 0.32 | 0.31 | 0.51 | 0.88 | 0.84 | 0.95 |
| | | LAVE(-) | All | F | All | 260 | 0.25 | 0.27 | 0.27 | 0.43 | 0.73 | 0.70 | 0.79 |
| Lym | 10^3 /μL | LAVE(-) | All | M | All | 200 | 1.15 | 1.28 | 1.23 | 2.01 | 3.55 | 3.26 | 3.79 |
| | | LAVE(-) | All | F | All | 260 | 1.04 | 1.16 | 1.13 | 1.78 | 3.17 | 2.91 | 3.33 |
| Neu% | % | LAVE(-) | All | MF | All | 485 | 36.5 | 39.9 | 38.8 | 54.1 | 70.1 | 68.9 | 71.8 |
| Eos% | % | LAVE(-) | All | M | All | 226 | 0.70 | 0.91 | 0.80 | 2.59 | 8.66 | 7.84 | 9.39 |
| | | LAVE(-) | All | F | All | 259 | 0.63 | 0.82 | 0.75 | 2.24 | 6.56 | 5.80 | 7.37 |
| Bas% | % | LAVE(-) | All | MF | All | 483 | 0.19 | 0.22 | 0.21 | 0.67 | 1.51 | 1.44 | 1.65 |

*(Continued)*

**Table 4.** (Continued)

| Item | Unit | LAVE | BMI | Sex | Age | N | LL_L | LL_H | LL | Me | UL | UL_L | UL_H |
|------|------|------|-----|-----|-----|---|------|------|-----|-----|-----|------|------|
| **Mon%** | % | LAVE(-) | All | M | All | 222 | 5.61 | 6.16 | **5.93** | **8.64** | **12.42** | 12.07 | 12.89 |
| | | LAVE(-) | All | F | All | 260 | 4.64 | 5.11 | **4.91** | **7.82** | **11.29** | 10.90 | 11.92 |
| **Lym%** | % | LAVE(-) | All | MF | All | 484 | 19.2 | 20.8 | **20.6** | **33.8** | **47.9** | 47.2 | 50.3 |
| **PLT** | 10^3 /μL | LAVE(-) | All | M | All | 222 | 108 | 135 | **132** | **205** | **288** | 279 | 302 |
| | | LAVE(-) | All | F | All | 261 | 120 | 142 | **135** | **227** | **330** | 318 | 339 |
| **MPV** | fL | LAVE(-) | All | MF | All | 487 | 7.35 | 7.50 | **7.41** | **9.14** | **11.8** | 11.64 | 12.03 |
| **PDW** | % | LAVE(-) | All | MF | All | 406 | 15.8 | 16.0 | **16.0** | **16.8** | **17.9** | 18.0 | 18.5 |

<To avoid separation, the narrower between-row spaces were set.>

damage, Hb levels are higher for higher BMI and lower EPO doses are required to maintain Hb levels in obese compared to normal weight patients [26]. In summary, the mechanism underlying the increased hematology parameters, including reticulocytes, in obesity is not fully known, and further investigations are necessary.

## 2. Comparison of Russian RIs with those reported by other countries

In **Table 4**, we summarized (some) RIs reported for hematology parameters and anemia-associated markers, as well as those provided by the manufacturer in the IFU. In the Turkish, Kenyan, Ghanian, Japanese, Saudi, Chinese and Indian studies [4–11], RIs were derived by the parametric method with the application of LAVE procedure according to the harmonized protocol provided by the IFCC C-RIDL. While in Canadian, Malaysian, African (Asmara), Moroccan, Omani, and Ethiopian studies, RIs were determined nonparametrically according to the CLSI guideline [27–33]. Additionally, we included hematology RIs commonly used across the Russian laboratories that are published in the Russian Methodological guidelines [34, 35], as well as the cutoff values that are provided in anemia-related guidelines [36–41].

For ease of comparison, the differences in the spans of RIs are shown by bar-charts in **Fig 4A and 4B** for representative parameters.

### 2.1 Erythrocyte and iron markers

The World Health Organization (WHO), the American Gastroenterological Association (AGA) and the Russian National Hematology Society (RNHS) define iron deficiency anemia as blood Hb levels below 130 g/L in men and below 120 g/L in nonpregnant women, and ferritin levels < 15 μg/L (WHO guidelines) or < 11.0 μg/L in addition to the related clinical symptoms (RNHS guidelines). AGA guidelines provide the recommendation in patients with anemia to use a cutoff of 45 ng/mL over 15 ng/mL when using ferritin to diagnose iron deficiency [36–38].

Based on the results of the current study, LL for Hb in males was slightly higher compared to the international guidelines listed above, and to the IFU (134 vs. 130 and 125 g/L, respectively), but comparable to other studies [4–9, 27–33]. The highest LLs and ULs for Hb and RBC were reported from Kenya and Ethiopia [7, 33]. This corresponds to Muckenthaler et al., who demonstrated that Hb, RBC and ferritin levels significantly rose with every 300-m increase in residential altitude, starting from as low as 300 m above the sea level [42] (Saint-Petersburg is located at the altitude of 3 m, vs 1795 m and 2355 m of Nairobi and Addis Ababa, respectively). Similar effects of high altitude were also reported in the Turkish multicenter study [6]. Omuse et al. also mentioned that the Kenyan study included urban

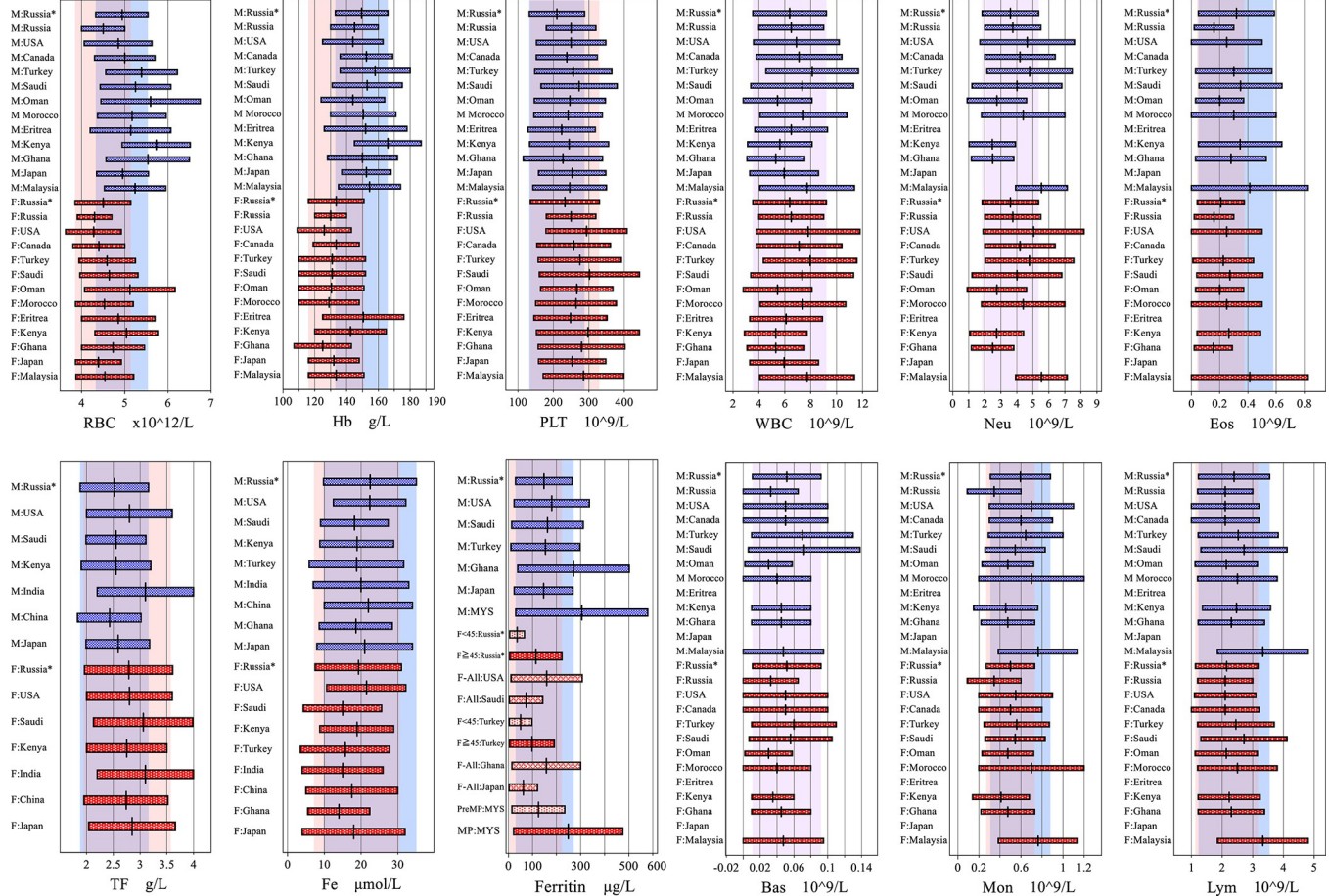

**Fig 4. A** Between-country comparison of RIs for erythrocyte and iron markers. **B** Between-country comparison of RIs for leukocyte parameters. Russia*—indicates RIs from the current Russian study; Russia (without *)–indicates RIs from the Russian Methodological Guidelines [33, 34]. The back shades indicate Russian RIs for male (blue), female (pink), and both genders (purple).

population only with sufficient nutrition and easy access to medical treatment. Therefore, their RI limits for Hb and RBC in both sexes are raised in comparison to other African studies (**Table 5 and Fig 4A and 4B**) [7]. Their interpretation of raised levels of erythrocyte parameters by nutrition is consistent with our findings of BMI-related raise of those RIs.

Ferritin LL in males (29.8 μg/L) in this study was much higher compared to the cutoffs provided by WHO, and RNHS guidelines (15, and 11 μg/L, respectively) [36, 38], to the Turkish study (13 μg/L) [6] and comparable to the LL in the IFU (23.9 μg/L) [18].

In Russian females we observed a slightly reduced LL of Hb in comparison to the AGA and RNHS guidelines (114 vs. 120 g/L) [37, 38], meanwhile significantly lower ferritin LL was observed in both age groups <45 and ≥45 years (5.5 and 5.3 vs. 45 and 11 μg/L, respectively) [36–38]. Even a lower level of ferritin was reported in the Turkish and Japanese studies (4.3 and 3.0 μg/L, respectively) [5, 6]. Low levels of ferritin in age group <45 years are associated with reduced iron storage due to menstrual bleeding, pregnancy or latent anemia. Bawua et al. showed age-related increase in ferritin and decrease in TFR, which was confirmed in this study [8]. However, LL for ferritin in Russian females for the age group >45 is still very low (5.3 μg/L). LL for Fe in Russian females (7.2 μmol/L) was also low compared to the IFU (10.7 μmol/L), but twice higher than those reported from Japan, Turkey, China, and India (4.0,

**Table 5. Comparison of RIs from this study with those from other studies.**

| Analyte | Unit | Sex | Russian study[a] | Beckman IFU[a] | Russian Methodological guidelines[d,e] [34, 35] | Kenyan study[f] [7] | Turkish study[g,h,i] [6] | Ghanian study[j] [8, 32] | Japanese study[a,o] [5] | Canadian study CALIPER[m] [27] | Malaysian study[a,o] [28] | Erithea (Asmara), Africa[g] [29] | Saudi Arabian study[l] [9] | Moroccan study[r] [30] | Ethiopian study[s] [33] | Omani study[i,t] [31] |
|---|---|---|---|---|---|---|---|---|---|---|---|---|---|---|---|---|
| RBC | ×10^6/μL | M | 4.46–5.46 | 4.06–5.63 | 4.0–5.0 | 4.94–6.52 | 4.43–6.07 | 4.57–6.5 | 4.35–5.55 | Age15-49: 4.3–5.7 Age50-79: 4.2–5.5 | <60yo: 4.53–5.95 >60yo: 3.86–5.62 | 4.2–6.07 | 4.56–6.22 | 4.37–5.96 | 4.26–6.68 | 4.45–6.75 |
|  |  | F | 3.88–4.94 | 3.63–4.92 | 3.9–4.7 | 4.31–5.76 | 3.96–5.31 | 4.0–5.46 | 3.86–4.92 | Age15-49: 3.8–5.0 | 3.87–5.21 | 4.0–5.7 | 3.94–5.25 | 3.86–5.2 | 4.02–6.15 | 4.07–6.17 |
|  |  | MF |  | 3.73–5.5 |  | 4.41–6.48 |  |  |  |  |  | 4.07–6.02 |  |  |  |  |
| Hb | g/L | M | 134–166 | 125–163 | 130–160 | 145–187 | 131–175 | 128–172 | 137–168 | 136–169 | <60yo: 135–174 | 126–178 | 136–180 | 130–171 | 121–188 | 124–164 |
|  |  | F | 114–148 | 109–143 | 120–140 | 120–165 | 110–152 | 107–143 | 116–148 | N/A | >60yo: 118–169 | 125–176 | 110–152 | 111–148 | 123–179 | 111–151 |
|  |  | MF |  | 114–159 |  | 128–190 |  |  |  |  | 116–151 | 126–177 |  |  |  |  |
| Ht | % | M | 40.1–48.3 | 36.7–47.1 | 40–48 | 43–55 | 39.2–52.2 | 39.4–52.1 | 41–50 | 40–50 | <60yo: 40.1–50.6 | 40.5–55.0 | 40.2–52.0 | 38.3–50 | 36.7–545 | 36–47 |
|  |  | F | 34.1–42.5 | 31.2–41.9 | 36–42 | 36–49 | 33.7–46.1 | 34.0–44.2 | 35–44 | N/A | >60yo: 35.7–48.9 | 37.9–52 | 33.6–44.5 | 33.5–43.9 | 36.9–51.59 | 33–43 |
|  |  | MF |  | 33.3–45.7 |  | 38–55 |  |  |  |  | 35.1–44.9 | 38.3–54.4 |  |  |  |  |
| MCV | fL | M | 82.3–96.5 | 73.0–96.2 | 80.0–100.0 | 76.5–95.5 | 77.2–95.7 | 71.9–97.4 | 83.6–98.2 | 82.5–98.0 | 80.6–95.5 | 85.7–100 | 75–94.4 | 77.4–94.2 | 74.8–93.94 | 62.5–88.5 |
|  |  | F | 82.5–94.2 | 75.5–95.3 |  | 73.4–95.8 |  |  |  | 82.5–98.0 |  | 85.5–100 |  | 75.1–94.7 | 77.3–98.82 |  |
|  |  | MF |  | 73.7–95.5 |  | 75.7–95.6 |  |  |  |  |  | 85.8–100 |  |  |  |  |
| MCH | pg | M | 28.2–32.7 | 23.8–33.4 | 27.0–31.0 | 25.1–32.8 | 25.2–32.2 | 23.2–31.8 | 27.5–33.2 | 27.6–33.3 | 26.9–32.3 | 28–33 | 25–32.5 | 25.2–32.3 | 24.86–32.8 | 20.81–31.2 |
|  |  | F |  | 24.7–32.8 |  | 24.4–32.7 |  |  |  |  |  | 26.5–32.6 |  | 24–32.3 | 26.3–33.58 |  |
|  |  | MF |  | 24.3–33.2 |  | 24.8–32.8 |  |  |  |  |  | 27.4–32.8 |  |  |  |  |
| MCHC | g/L | M | 330–353 | 325–363 | 300.0–380.0 | 324–354 | 319 -350[g] | 290–352 | 317–353 | 325–352 | 319–353 | 304–337 | 314–361 | 317–360 | 320.6–365 | 31–37.2 |
|  |  | F |  | 323–356 |  | 320–350 |  |  |  |  |  | 300–337 |  | 312–360 | 320.0–360 |  |
|  |  | MF |  | 325–358 |  | 322–352 |  |  |  |  |  | 302–338 |  |  |  |  |
| RDW-SD | fL | M | 37.7–45.5 | 36.5–45.9 |  |  |  |  |  |  | 37.5–48.1 |  |  |  |  |  |
|  |  | F |  | 37.6–50.3 |  |  |  |  |  |  |  |  |  |  |  |  |
|  |  | MF |  | 37.1–47.8 |  |  |  |  |  |  |  |  |  |  |  |  |
| RDW-CV | % | M | 12.2–14.2 | 12.1–16.2 | 11.5–14.5 | 11.3–14.7 | 12.2–16.3[g,h] | 11.7–16.0 |  | 11.4–13.5 | 12–14.8 | 12.3–15.5 | 11.3–14.1 |  | 12.46–17.56 | 11.1–17.8 |
|  |  | F | 12.3–15.0 | 12.3–17.7 |  | 11.4–15.8 |  | 12.0–17.3 |  | 11.4–13.5 |  | 12.2–17 | 11.0–16.4 |  | 12.4–15.59 |  |
|  |  | MF |  | 12.3–17.0 |  | 11.3–15.2 |  |  |  |  |  | 12.3–15.6 |  |  |  |  |
| Ret | ×10^9/L | M | 23.0–84.1 | 18.8–108.6 | 10.0–90.0e |  |  |  |  |  |  |  |  |  |  |  |
|  |  | F |  | 23.0–93.5 | 10.0–90.0e |  |  |  |  |  |  |  |  |  |  |  |
|  |  | MF |  | 22.1–96.3 |  |  |  |  |  |  | 0.4–1.6 |  |  |  |  |  |
| Ret% | % | M | 0.54–1.77 | 0.42–2.23 | 0.24–1.67 |  |  |  |  |  |  |  |  |  |  |  |
|  |  | F |  | 0.51–2.17 | 0.12–2.05e |  |  |  |  |  |  |  |  |  |  |  |
|  |  | MF |  | 0.5–2.17 | 0.2–1.2e |  |  |  |  |  |  |  |  |  |  |  |

(Continued)

**Table 5.** (Continued)

| Analyte | Unit | Sex | Russian study[a] | Beckman IFU[a] | Russian Methodological guidelines[d,e] [34, 35] | Kenyan study[f] [7] | Turkish study[g,h,i] [6] | Ghanian study[j] [8, 32] | Japanese study[a,o] [5] | Canadian study CALIPER[m] [27] | Malaysian study[a,o] [28] | Erithea (Asmara), Africa[g] [29] | Saudi Arabian study[j] [9] | Moroccan study[r] [30] | Ethiopian study[s] [33] | Omani study[i,t] [31] |
|---|---|---|---|---|---|---|---|---|---|---|---|---|---|---|---|---|
| MRV | fl | M | | 97.5–122.7 | 101.1–128.8 | | | | | | | | | | | |
| | | F | | 96.4–118 | 101.4–125.0[e] | | | | | | | | | | | |
| | | MF | 97.6–117.8 | 97.4–120.2 | | | | | | | 93.9–124.1 | | | | | |
| IRF | % | M | | 30–54 | 19–41[e] | | | | | | | | | | | |
| | | F | | 26–52 | 17–40[e] | | | | | | | | | | | |
| | | MF | 24.5–49.8 | 29–53 | | | | | | | | | | | | |
| WBC | ×10³/µL | M | | 3.6–10.2 | | 3.13–8.1 | | | | 3.8–10.4 | | 3.7–9.3 | | 4.1–10.8 | 3.31–11.62 | |
| | | F | | 3.8–11.8 | | 2.89–7.72 | | | | 3.8–10.4 | | 3.3–8.9 | | 4.1–10.7 | 3.24–10.05 | |
| | | MF | 3.54–9.21 | 3.6–11.2 | 4.0–9.0 | 3.08–7.83 | 4.39–11.59 | 3.08–7.53 | 3.3–8.6 | | 4.08–11.37 | 3.4–9.0 | 3.38–11.31 | | | 2.79–8.09 |
| Neu | ×10³/µL | M | | 1.7–7.6 | | 1.02–3.92 | | | | Age17-50:1.8–7.2 Age51-79:2.0–6.4 | | | | 1.8–7 | 1.01–7.22 | |
| | | F | | 1.9–8.2 | | 1.07–4.42 | | | | Age17-50:2.0–7.4 Age51-79:2.0–6.4 | | | | 1.8–7 | 1.08–6.69 | |
| | | MF | 1.67–5.85 | 1.8–7.8 | 2.0–5.5 | 1.05–4.08 | 2.04–7.54 | 1.17–3.81 | | | 3.93–7.15 | | 1.21–6.81 | | | 0.91–4.61 |
| Eos | ×10³/µL | M | 0.05–0.59 | 0.0–0.5 | | 0.05–0.64 | | | | 0.1–0.2 | | | 0.05–0.64 | 0–0.6 | 0.05–1.21 | |
| | | F | 0.041–0.38 | 0.0–0.5 | | 0.04–0.49 | | | | 0.1–0.2 | | | 0.04–0.51 | 0–0.5 | 0.04–1.12 | |
| | | MF | | 0–0.5 | 0.02–0.3 | 0.04–0.59 | 0.02–0.5 | 0.02–0.29 | | | 0–0.827 | | | | | 0.03–0.37 |
| Bas | ×10³/µL | M | | 0.0–0.1 | | 0.01–0.08 | | | | 0.0–0.1 | | | 0.01–0.14 | 0–0.08 | 0.01–0.05 | |
| | | F | | 0.0–0.1 | | 0.01–0.06 | | | | 0.0–0.1 | | | 0.01–0.11 | 0–0.08 | 0.0–0.05 | |
| | | MF | 0.011–0.092 | 0.0–0.1 | 0.0–0.065 | 0.01–0.07 | 0.0–0.09[g] | 0.01–0.08 | | | 0–0.095 | | | | | 0.002–0.058 |
| Mon | ×10³/µL | M | 0.31–0.88 | 0.3–1.1 | | 0.15–0.76 | | | | Age6-44:0.2–0.8 Age45-79:0.3–0.9 | | | | 0.2–1.2 | 0.24–0.88 | |
| | | F | 0.27–0.73 | 0.2–0.9 | | 0.14–0.68 | | | | Age6-44:0.2–0.8 Age45-79:0.2–0.8 | | | | 0.2–1.2 | 0.27–0.87 | |
| | | MF | | 0.3–1.0 | 0.09–0.6 | 0.14–0.74 | 0.26–0.94 | 0.22–0.73 | | | 0.39–1.14 | | 0.26–0.83 | | | 0.23–0.72 |
| Lym | ×10³/µL | M | 1.23–3.55 | 1.0–3.2 | | 1.36–3.58 | | | | 1.0–3.2 | | | | 1.2–3.8 | 1.1–3.84 | |
| | | F | 1.13–3.17 | 1.1–3.1 | | 1.22–3.24 | | | | 1.0–3.2 | | | | 1.2–3.8 | 1.2–3.98 | |
| | | MF | | 1.0–3.0 | 1.2–3.0 | 1.29–3.4 | 1.21–3.77 | 1.22–3.38 | | | 1.85–4.81 | | 1.31–4.12 | | | 1.12–3.15 |

(Continued)

**Table 5.** (Continued)

| Analyte | Unit | Sex | Russian study[a] | Beckman IFU[a] | Russian Methodological guidelines[d,e] [34, 35] | Kenyan study[f] [7] | Turkish study[g,h,i] [6] | Ghanian study[j] [8, 32] | Japanese study[a,o] [5] | Canadian study CALIPER[m] [27] | Malaysian study[a,o] [28] | Erithea (Asmara), Africa[g] [29] | Saudi Arabian study[j] [9] | Moroccan study[r] [30] | Ethiopian study[s] [33] | Omani study[i,t] [31] |
|---|---|---|---|---|---|---|---|---|---|---|---|---|---|---|---|---|
| Neu% | % | M | **38.8–70.1** | 43.5–73.5 | 47.0–72.0 | 27.4–60.3 | 40–74 | 29–62 | | | | | 27.4–68.5 | | | |
| | | F | | 42.7–76.8 | | 29.5–65.4 | | | | | | | | | | |
| | | MF | | 43.3–76.6 | | 28.0–63.3 | | | | | | | | | | |
| Eos% | % | M | **0.8–8.66** | 0.8–8.1 | 0.5–5.0 | 1.2–11.8 | 0–6 | 0.5–10.3 | | | | | 0.89–8.54 | | | |
| | | F | **0.75–6.56** | 0.5–7.0 | | 0.8–9.4 | | 0.4–6.5 | | | | | 0.52–7.21 | | | |
| | | MF | | 0.6–7.9 | | 1.1–11.9 | | | | | | | | | | |
| Bas% | % | M | **0.21–1.51** | 0.2–1.5 | 0.1–1.0 | 0.4–1.2 | 0.1–1.0[g] | 0.2–1.5 | | | | | 0.09–1.59 | | | |
| | | F | | 0.2–1.3 | | 0.3–1.0 | | | | | | | | | | |
| | | MF | | 0.2–1.4 | | 0.3–1.0 | | | | | | | | | | |
| Mon% | % | M | **5.93–12.42** | 5.5–13.7 | 3.0–11.0 | 3.5–14.3 | 4.0–12 | 5.0–13.7 | | | 3.1–11.6 | | | | | |
| | | F | **4.91–11.29** | 4.3–10.9 | | 3.2–11.0 | | | | | 3–11.8 | | | | | |
| | | MF | | 4.5–12.5 | | 3.4–13.3 | | | | | 3.1–11.7 | | | | | |
| Lym% | % | M | **20.6–47.9** | 15.2–43.3 | 19.0–37.0 | 28.2–60.3 | 17–47 | 27–60 | | | 22–59.9 | | 23–58.4 | | | |
| | | F | | 16.0–45.9 | | 25.5–59.3 | | | | | 22.3–58.2 | | | | | |
| | | F | | 16.0–43.5 | | 27.2–60.0 | | | | | 22–59.2 | | | | | |
| PLT | ×10³/µL | M | **132–288** | 152–348 | 180–320 | 133–356 | 152–383 | 115–339 | 158–348 | Age14–26: 139–320 Age27–79: 152–324 | 142–350 | 128–318 | 165–380 | 145–338 | 164–403 | 146–347 |
| | | F | **135–330** | 179–408 | | 152–443 | | 157–402 | | Age14–26: 158–362 Age27–79: 153–361 | 171–399 | 145–352 | 160–443 | 150–378 | 202–444.5 | 164–368 |
| | | MF | | 159–386 | | 144–409 | | | | | | 134–344 | | | | |
| MPV | fL | M | **7.41–11.8** | 7.4–11.4 | 7.4–10.4 | | 7.0–11.8[g] | | | 7.0–10.3 | 8.9–11.9 | | 6.36–10.0 | 9.4–13.7 | | 6.61–10.3 |
| | | F | | 7.9–10.8 | | | | | | 7.0–10.3 | | | 6.34–11.0 | 9.0–13.7 | | |
| | | MF | | 7.5–11.2 | | | | | | | | | | | | |
| PDW | % | MF | **16.0–17.9** | | | | | | | | | | 14.9–17.9 | | | 7.7–17.4 |

3.5, 5.0 and 4.0 μmol/L, respectively) (**Table 5** and **Fig 4B**). The discrepancy is attributable to the fact, that those studies did not refer to ferritin and CBC results while establishing RIs for Fe. In any case, these findings indicate a high prevalence of latent IDA among Russian population, as was shown in the WHO report (21.1% prevalence of IDA among non-pregnant Russian women of reproductive age) [43] (Table 6).

**2.2 Platelet counts.** LLs of PLT in the current Russian study (male: 132; female: 135×10$^3$/μL) were significantly lower compared to the LLs shown in the IFU (152; 179×10$^3$/μL) [44] and LLs in the Russian Methodological guidelines for interpretation of blood tests (180×10$^3$/μL for both sexes) [34]. In fact, the Russian RIs for PLT were the lowest among all countries examined, as shown in **Table 5** [4–11, 27–33] and **Fig 4**. Although the PLT RI from the Russian Methodological guidelines was recently revised to 150–400 ×10$^3$/μL, RIs from this study are still shifted to a lower side: 132–288 (male) and 135–330 (female) ×10$^3$/μL. The reason for this difference is unknown. Hence it certainly requires further investigation by inclusion of more volunteers from Caucasian population. Noticeably, that in most studies female RIs for PLT are shifted to a higher side, which can be explained by the female predominance of iron deficiency, which tends to cause increase of thrombopoiesis [7, 45, 46].

**2.3 Vitamin B12 and folate.** The recommended laboratory evaluation for patients with suspected macrocytic anemia includes CBC, VitB12, and folate levels assessment. The concentration of plasma vitamin B12 suggested by WHO for defining vitamin B12 deficiency is < 150 pmol/L, whereas the American Academy of Family Physicians (AAFP) and RNHS suggest <111 and <103 pmol/L respectively, which corresponds to the cutoff provided in the IFU (107 pmol/L) [39–41, 47]. Our LLs for VitB12 (male 113 and female 125 pmol/L) are above the AAFP, and RNHS cutoffs, and the LL reported in the Japanese study (91 and 112 pmol/L for males and females respectively), but noticeably lower than the WHO cutoff and the IFU LL of a "normal range" (133 pmol/L) (**Table 4** and **Fig 4A**) [5, 39, 47].

Folate LLs of Russian males and females (3.15 and 3.58 ng/mL) were a little lower than those recommended by WHO (< 4 ng/mL) and provided in the IFU for the USA population (5.9 ng/mL), but comparable to the IFU cutoff derived for the British population (3.1 ng/mL), and again much lower than the LLs reported in the Japanese study for males and females (8.7 and 10.8 ng/mL, respectively), that could be a result of an insufficient dietary intake (**Table 4** and **Fig 4B**) [5, 39, 48].

**2.4 Leukocyte counts.** LL of RI obtained in the current study for both sexes was lower vs. those provided in Russian Methodological guidelines (3.5 vs. 4.0 x10$^3$/μL) [34].

Significant between-country differences were observed for RI limits of Neu count, with LLs being conspicuously low in Kenya, Ghana, Oman, and Ethiopia [7, 8, 31, 33]. For black African population, Omuse et al. speculated that the low RIs for Neu count may be associated with the DARC-null genotype, an evolutionary adaptation that is thought to make Africans less susceptible to Plasmodium vivax infections [7]. Another factor could be benign ethnic neutropenia, a common form of neutropenia worldwide that affects African and Middle Eastern ethnicities, especially people living at high altitude [49].

RIs limits for both sexes for Eos in the Russian study were comparable with limits from Kenya, Ghana, Saudi Arabia, Morocco, Omani and Ethiopia, and with those provided in the IFU [7–9, 30, 31, 33, 44]. Omuse et al., observed much lower RIs for Eos in comparison to other African studies, and attributed it to the recruitment of the urban population with a lower risk of parasitic infection [7].

The UL for Mon counts in Russia was prominently low, but comparable to RIs of the Russian Methodological guidelines [34], and to those of the Kenyan, Ghanian, and Omani studies [7, 8, 31].

**Table 6. Comparison of RIs from this study with those from other studies.**

| Analyte | Unit | Sex | Russian study[b,c] | Beckman IFU[b,c] | Kenyan study[e] [7] | Turkish study [6] | Ghanian study[k] [8, 32] | Asian study[c, l] [4, 5] | Chinese study[b] [10] | Indian study[u] [11] | Saudi Arabian study[p] [9] |
|---|---|---|---|---|---|---|---|---|---|---|---|
| Fe | µmol/L | M | 9.8–35.1 | 12.5–32.2 | | 5.9–31.6 | 8.6–28.5 | 8.0–34.0[h,6] | 10.0–34.0 | 7.0–33.0 | 9.0–27.4 |
| | | F | 7.2–30.5 | 10.7–32.2 | | 3.5–27.8 | 5.5–22.4 | 4.0–32.0 | 5.0–30.0 | 4.0–26.0 | 4.3–25.6 |
| | | MF | | | 8.8–28.9 | | | 5.0–33.0 | | | |
| UIBC | µmol/L | M | 18.7–54.1 | | | 21.5–64.7 | | 19–556 | | | 23.6–51.2 |
| | | F | 26.2–63.5 | 27.8–63.6 | | 28.3–78.1 | | 24–75 | | | 29.2–71.9 |
| | | MF | | | | | | 21–69 | | | |
| TIBC | µmol/L | M | 46.0–71.4 | N/A | | 44.0–82.2 | | | | | |
| | | F | 47.0–78.4 | | | 46.8–88.9 | | | | | |
| TF | g/L | M | 1.88–3.16 | | 1.9–3.2 | | | 1.99–3.18 | 1.83–3.02 | | 1.99–3.11 |
| | | F | 1.94–3.57 | 2.0–3.6 | 2.0–3.5 | | | 2.04–3.66 | 1.95–3.52 | | 2.13–3.99 |
| | | MF | | | | | | 2.01–3.52 | | | |
| TFRSat | % | M | 16.6–64.3 | N/A | 15–49 | | | | | | |
| | | F | 11.5–51.3 | | 10.0–44 | | | | | | |
| Ferr | µg/L | M | 29.8–271[c] | 23.9–336.2 | | 13.0–276.0 | | 24-268[i] | | | |
| | | F | <45 yo: 5.5–67 >45 yo: 5.3–223 | 11. 0–306.8 | | <45 yo: 4.3–91 >45 yo: 5.9–175 | | 3–121 | | | |
| VitB12 | pmol/L | M | 113-549[c] | 133–675[1] | | | | 91–520[6] | | | |
| | | F | 125–710 | 107–133[2] | | | | 112–606 | | | |
| | | MF | | <107[3] | | | | 100–575 | | | |
| Folate | ng/mL | M | 3.15–16.5[c] | | | | | 8.7–34.7[i] | | | |
| | | F | 3.58–21.1 | 5.9 - >24.8[4] | | | | 10.8–46 | | | |
| | | MF | | 3.1–19.9[5] | | | | 9.4–40 | | | |

1 -Normal range; 2 -Indeterminate Range; 3—Deficiency range; 4—USA population; 5—British population; 6—all countries from Asian study

a) UniCel DxH 800 (Beckman Coulter Inc., USA); b) AU 5800 (Beckman Coulter Inc., USA); c) UniCel DxI (Beckman Coulter Inc., USA); d) Manual method (Goryaev's chamber); e) Gen-S (Beckman Coulter Inc., USA); f) AST 5 Diff (Beckman Coulter Inc., USA); g) LH 780 (Beckman Coulter Inc, USA); h) XT-2000i (Sysmex Corporation, Japan); i) Cell-Dyn 3700 Sapphire (Abbott Laboratories, USA); j) XN 1000 (Sysmex Corporation, Japan); k) AU 480 (Beckman Coulter Inc., USA); l) UniCel DxC (Beckman Coulter Inc.); m) HmX analyzer (Beckman Coulter Inc.); o) XE 5000 (Sysmex Corporation, USA); p) Architect 16000c (Abbott Laboratories, USA); r) KX21N (Sysmex Corporation, Japan); s) XS 500i (Sysmex Corporation, Japan); t) XS-1000i (Sysmex Corporation, Japan); u) DxC 800 (Beckman Coulter Inc., USA)

Meanwhile, RI limits for Lym count are comparable across countries except Malaysia, which showed elevated RIs for WBC and all differential counts [28] of unknown epidemiological background.

## Limitations

The methodologies used in this study were in close alignment with internationally harmonized protocol elaborated by IFCC C-RIDL in 2012. Nevertheless, there are two limitations due to our recruitment scheme that may impact the ability to generalize our findings to the overall population. The first limitation was due to our adoption of convenient sampling targeting individuals who are easily accessible from clinical laboratories, like hospital personnels and their family, and anyone who could access the study-advertisement posted in the hospital. Therefore, unlike population-based random sampling recommended for epidemiological studies of public health interest, our sample population may not reflect individuals who experience different living environments and physical activities. Nevertheless, it would not have been

practical to set up such a large-scale population-based sampling just for the derivation of RIs. In fact, recent RI studies conducted a nationwide scale in Japan [50], China [10], and Turkey [51], which basically adopted the C-RIDL scheme, reported no regional differences within each country in any of the biochemical analytes examined. Therefore, it is generally agreed upon that so long as lifestyle, physical activity level and living environment are nearly comparable, the convenient sampling is acceptable for determination of RIs.

The second limitation is the adoption of very lenient exclusion and inclusion criteria in order to attract sufficient numbers of volunteers to participate. However, we made efforts to identify inappropriate individuals after measurements: i.e., restriction of individuals with obesity in determining RIs for analytes with BMI related changes, and exclusion of those exhibiting iron-deficiency state detected by the LAVE method. Another limitation of this study is somewhat arbitrary nature of setting conditions for secondary exclusion using the LAVE method. In fact, there can be many options to attain optimal LAVE effect without much data-size reduction, which depend on how many reference tests to set up and how much to expand the RI limits in judging test result abnormality. Although we adopted 3% expansion on both-end of the RI from our past trial and error experiences, other settings can lead to similar results.

## Conclusion

With the participation of 506 healthy Russian adults, we investigated sources of variation and determined RIs parametrically for hematology parameters, including related iron markers and vitamins. Gender-difference requiring separate RIs for each gender were apparent in most of the analytes, while age-related changes in RVs were noted only for ferritin in females. BMI-related increases in RVs were observed prominently for reticulocyte parameters, which required exclusion of individuals with BMI$>$28 kg/m$^2$ before determining the RIs. The LAVE method was found effective in excluding individuals with latent anemia without inclusion of iron makers in the reference tests.

International comparison revealed that Russian RIs featured a lower side shift of platelet counts. Besides, as in African countries, Russian RIs for total leukocyte and neutrophil counts were lower compared to most of other countries. The observed ethnic differences will require further investigation for validation.

## Supporting information

**S1 Fig. Bar-chart comparison of LAVE effect on RIs for RBC and AAM.** The RIs were derived in four ways: by parametric (P) or non-parametric (NP) method with or without application of latent abnormal values exclusion method (LAVE) method. Each horizontal bar represents the RI, and the vertical line in the center corresponds to the midpoint. The shades on both ends of the bar represent 90% CI for the limits of the RI predicted by the bootstrap method. The RIs derived for twenty parameters related to red blood cells (RBC) and anemia associated markers (AAM) for males (M: blue) and females (F: red) are shown in this figure. (PDF)

**S1 File.**
(XLSX)

**S2 File.**
(XLSX)

## Acknowledgments

The authors express sincere gratitude to Beckman Coulter, LLC (Russia) for their generous support of the assay reagents. We are grateful to Yury Andreychuk, CEO of Helix Laboratories Services, and the Helix staff for their kind assistance in the recruitment of volunteers, sample preparations, and provision of sampling equipment.

## Author Contributions

**Conceptualization:** Kiyoshi Ichihara.

**Data curation:** Anna Ruzhanskaya, Kiyoshi Ichihara, Anton Vasiliev, Ekaterina Vilenskaya.

**Formal analysis:** Anna Ruzhanskaya, Kiyoshi Ichihara.

**Investigation:** Anna Ruzhanskaya, Irina Skibo, Dmitry Butlitski, Anton Vasiliev, Galina Agarkova.

**Methodology:** Kiyoshi Ichihara.

**Project administration:** Anna Ruzhanskaya, Irina Skibo, Nina Vybornova.

**Resources:** Anna Ruzhanskaya, Irina Skibo, Nina Vybornova.

**Software:** Kiyoshi Ichihara.

**Supervision:** Anna Ruzhanskaya, Kiyoshi Ichihara, Nina Vybornova.

**Validation:** Anna Ruzhanskaya, Kiyoshi Ichihara, Elena Sukhacheva.

**Visualization:** Anna Ruzhanskaya, Kiyoshi Ichihara, Elena Sukhacheva.

**Writing – original draft:** Anna Ruzhanskaya, Kiyoshi Ichihara.

**Writing – review & editing:** Anna Ruzhanskaya, Kiyoshi Ichihara, Elena Sukhacheva, Irina Skibo, Nina Vybornova, Dmitry Butlitski, Anton Vasiliev, Galina Agarkova, Ekaterina Vilenskaya, Vladimir Emanuel, Svetlana Lugovskaya.

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
