## [Decision Letter · Decision Letter 0]

27 Oct 2023

PONE-D-23-22704Derivation of Russian-specific reference intervals for complete blood count, iron markers and related vitaminsPLOS ONE

Dear Dr. Ichihara,

Thank you for submitting your manuscript to PLOS ONE. After careful consideration, we feel that it has merit but does not fully meet PLOS ONE’s publication criteria as it currently stands. Therefore, we invite you to submit a revised version of the manuscript that addresses the points raised during the review process.

We look forward to receiving your revised manuscript.

Kind regards,

Theresa Ukamaka Nwagha, M.B.B.S., M.P.H., FMCPath, M.Sc., M.D.,

Academic Editor

PLOS ONE

3. We notice that your supplementary tables are included in the manuscript file. Please remove them and upload them with the file type 'Supporting Information'. Please ensure that each Supporting Information file has a legend listed in the manuscript after the references list.

4. We notice that your supplementary figures are uploaded with the file type 'Figure'. Please amend the file type to 'Supporting Information'. Please ensure that each Supporting Information file has a legend listed in the manuscript after the references list.

Additional Editor Comments:

The reviewers has returned their reviews with comments. Please go through and address issues raised adequately

Reviewers' comments:

Reviewer's Responses to Questions

**Comments to the Author**

1. Is the manuscript technically sound, and do the data support the conclusions?

Reviewer #1: Yes

Reviewer #2: No

Reviewer #3: Yes

2. Has the statistical analysis been performed appropriately and rigorously? 

Reviewer #1: Yes

Reviewer #2: No

Reviewer #3: I Don't Know

3. Have the authors made all data underlying the findings in their manuscript fully available?

Reviewer #1: Yes

Reviewer #2: No

Reviewer #3: Yes

4. Is the manuscript presented in an intelligible fashion and written in standard English?

Reviewer #1: Yes

Reviewer #2: No

Reviewer #3: Yes

5. Review Comments to the Author

Reviewer #1: Data on the RIs when iron deficiency is excluded would be useful. This is said to be shown in a figure but I cannot see the figure. A Table similar to Table 3 would be informative. The differences from other published ranges are interesting and informative.

Reviewer #2: The methodology for subject recruitment was not described adequately. The methodology needs to be written understandably. Avoid quoting other reference manuals as part of your method, but make efforts to describe your methodology thoroughly. I also noticed that your normal subjects included smokers and alcohol consumers. Did you compare the smokers versus non-smokers in your reference range calculation? Did you assess if the alcohol consumption affected the reference range calculation? On what basis have you decided to include smoking and alcohol consumption?

Also, the results are scattered without proper directions. It would be best to amend your methodology to reflect your result section accurately. There is no synchronization in the findings and method.

This paper also requires major English language editing. It is very difficult to understand the content in the current language.

A major revision is required; otherwise, i suggest rejecting this manuscript completely.

Reviewer #3: The study is important as it define RIs of hematological parameters including related iron markers and vitamins for adult population in Russian in accordance with the harmonized protocol and explore their specific features from an international perspective. This will help avoid of diagnostic errors, allow clinicians to interpret with greater specificity the hematological examinations and to improve the quality of medical care distributed to patients.

Here Some comments may need to clear

-What are the limitations of the study?

- Can the authors consider the selection of study participants based on their willingness to participate (with no specific randomization method)? ...is the main limitation of this study?

- Does the results indicated that the currently used reference interval does not represent the population in Russian.

- The author should maintain a single format in all references (---year, volume, issue, page)—accordingly recheck references no. 41,44.

- Re-check the published date in year in References No.2.

- Recheck reference no. 4, 13 regarding the date of publication in day , month

- Recheck the published date in years in References No.20, and issue.

6. PLOS authors have the option to publish the peer review history of their article (what does this mean?). If published, this will include your full peer review and any attached files.

Reviewer #1: No

Reviewer #2: No

Reviewer #3: No

---

## [Author Response · Author response to Decision Letter 0]

15 Dec 2023

 Pleaase refer to our responses in color at Page 94−102 of this file 

Reviewer #1: Data on the RIs when iron deficiency is excluded would be useful. This is said to be shown in a figure but I cannot see the figure. A Table similar to Table 3 would be informative. The differences from other published ranges are interesting and informative.

Our responses→Thanks for your time and efforts for reviewing our manuscript. 

Data on the RIs when iron deficiency was excluded were listed in Table 3. As for the figure, it was clearly indicated in Figure 3. In the figure, we realized that amongst the three methods of excluding iron deficiency, it was difficult to understand which one we finally adopted. Therefore, we updated the figure by emphasizing the second method (shown in the 4th row) was the one we adopted by using dark-green shade and RI limits in red.<** Please refer Page 97 of this PDF file **>

As for the last sentence of the comments, we appreciate your kind message regarding our international comparison of RI limits recent reported, which we laid out in Table 4 and Fig. 4.

=========>

Reviewer #2: The methodology for subject recruitment was not described adequately. The methodology needs to be written understandably. Avoid quoting other reference manuals as part of your method, but make efforts to describe your methodology thoroughly. I also noticed that your normal subjects included smokers and alcohol consumers. Did you compare the smokers versus non-smokers in your reference range calculation? Did you assess if the alcohol consumption affected the reference range calculation? On what basis have you decided to include smoking and alcohol consumption?

Our responses → We really appreciate your time and efforts for reviewing our manuscript and for providing us critical comments to improve our manuscript. 

We apologize for the lack of sufficient descriptions on our methodology, especially for subject recruitment. We have added the descriptions extensively as red-marked in the revised manuscript. 

Regarding the influence of smoking and drinking habits, we analyzed their associations with test results by calculating “partial correlation coefficient (rp)” using multiple regression analysis and listed them in Table 1, which was shown in Page 98 of this PDF file for your quick reference. 

As we regarded, |rp|>0.30 as significant, smoking and drinking habits were deemed to not have influence on RVs, while BMI and age did influence RVs, for which we managed by restricting BMI at 28kg/m2, and by partitioning by age if necessary.

Regarding the inclusion of individuals with drinking and smoking habits, we followed the consensus reached among IFCC delegates regarding the protocol to “not exclude anyone who considered themselves as healthy” so long as the exclusion criteria were fulfilled. The lenient recruitment policy was adopted to facilitate recruitment, due to the difficulty in identifying “truly healthy individuals”. However, it certainly entailed the need for secondary exclusion of inappropriate individuals after the measurements, as we emphasized the issue in the revised Method section. 

Also, the results are scattered without proper directions. It would be best to amend your methodology to reflect your result section accurately. There is no synchronization in the findings and method.

Our response → Thanks for the advice. In Methods, there were three subsections in “4) Statistical methods”.

4.1) Sources of variation of reference values

4.2) Criteria for partitioning RVs.

4.3) Derivation of reference intervals 

These three sub sections have been matched to the three subheadings of the revised “Results” under the same name to ensure synchronization between the findings and methods, as suggested by the reviewer. 

This paper also requires major English language editing. It is very difficult to understand the content in the current language.

Our responses → Thank you for pointing out the problem. The revised manuscript was thoroughly checked and edited by native English speaker with medical and scientific background.

A major revision is required; otherwise, i suggest rejecting this manuscript completely.

Our responses → We have made all changes as requested and addressed all comments. 

=========>

Reviewer #3: The study is important as it define RIs of hematological parameters including related iron markers and vitamins for adult population in Russian in accordance with the harmonized protocol and explore their specific features from an international perspective. This will help avoid of diagnostic errors, allow clinicians to interpret with greater specificity the hematological examinations and to improve the quality of medical care distributed to patients.

Our responses → We really appreciate your time and efforts for reviewing our manuscript and providing us comments to improve our manuscript. We revised the manuscript as described below. 

Here Some comments may need to clear

-What are the limitations of the study?

- Can the authors consider the selection of study participants based on their willingness to participate (with no specific randomization method)? ...is the main limitation of this study? 

Our responses → We understand that this comment raises the issue of appropriateness of our sampling scheme in recruiting healthy volunteers. To address this issue, we set up a new subheading of “Limitations” at the end of Discussion and expressed our thoughts on the issue as shown below. 

“Limitations

 The methodologies used in this study were in close alignment with internationally harmonized protocol elaborated by IFCC C-RIDL in 2012. Nevertheless, there are two limitations to the generalizability of our findings due to our recruitment scheme. One of the limitation was brought about by our adoption of convenient sampling targeting individuals who are easily accessible from clinical laboratories, like hospital personnels and their family, and anyone who could get access to the study-advertisement posted in the hospital. Therefore, unlike population-based random sampling recommended for epidemiological studies of public health interest, our sample population may not reflect individuals in quite different living environments and physical activities. Nevertheless, it is not practical to set up such a grand-scale population-based sampling just for deriving RIs. In fact, recent RI studies conducted in a nationwide scale in Japan [50], China [10], and Turkey[51], which basically adopted the C-RIDL scheme, reported no regional differences within each country in any of the biochemical analytes examined. Therefore, it is generally agreed upon that so long as life style, physical activity level and living environment are not biased, the convenient sampling is acceptable for determination of RIs. 

Another limitation regarding the generalizability of our finding is the adoption of very lenient exclusion and inclusion criteria to attract sufficient number volunteers to participate. However, we made efforts to identify inappropriate individuals after measurements: i.e., restriction of individuals with obesity in determining RIs for analytes with BMI related changes, and exclusion of those exhibiting iron-deficiency state detected by the LAVE method.” 

- Does the results indicated that the currently used reference interval does not represent the population in Russian.

Our responses →Thank you very much for pointing out this important question. For majority of the parameters, we didn’t find any significant differences of the RIs derived in the current study compared to the RIs in the manufacturer’s IFU, and RIs from Russian Methodological Recommendations (MR). However, prominent differences were observed in LL of RIs for Hb, Fe, Ferritin in females, which are meant to be used in diagnosing iron deficiency anemia (IDA). Their LLs of this study was much lower than that of MR, and also lower than the IDA cut-off value presented in Clinical Guidelines (AGA, WHO, local) and IFU (except Hb). LLs of our study for WBC, PLT, and UL for Neu in both sexes were lower than those of MR. 

A potential limitation of the frequently used RIs from Russian Methodological Recommendations is that they were established a long time ago (more than 30 years for some parameters) and some of the analytical systems used in that analysis remain unknown.

In addition, further studies (e.g. clinical validation) would aid in subsequent validation of our findings, which remain the first set of RIs for the Russian population using up-to-date methods proposed by C-RIDL. 

- The author should maintain a single format in all references (---year, volume, issue, page)—accordingly recheck references no. 41,44.

- Re-check the published date in year in References No.2.

- Recheck reference no. 4, 13 regarding the date of publication in day, month

- Recheck the published date in years in References No.20, and issue.

Our responses →Thank you for highlighting errors in the References. We have amended them accordingly, together with other typos and inconsistent formatting.

---

## [Decision Letter · Decision Letter 1]

9 Apr 2024

PONE-D-23-22704R1Derivation of Russian-specific reference intervals for complete blood count, iron markers and related vitaminsPLOS ONE

Dear Dr. Ichihara,  After careful consideration of your interesting manuscript, we feel that it has merit but does not fully meet PLOS ONE’s publication criteria as it currently stands. Therefore, we invite you to submit a minor revised version of the manuscript that addresses the points raised during the review process.

**ACADEMIC EDITOR: ****Dr. Ichihara****This is an interesting manuscirpt that show relevant informations on reference values of some biochemilca parameters. I reccomend to evaluate the suggestion of the reviewer 5 "**I suggest that the authors provide more details about the secondary patient exclusion criteria (information about health status acquired in the questionnaire). It was not clear how the exclusion and inclusion, even presenting a reference to Cohen, required more details". Such information is relevant to strenghten your results. Our decision is justified on PLOS ONE’s publication criteria and not, for example, on novelty or perceived impact.

We look forward to receiving your revised manuscript.

Kind regards,

José Luiz Fernandes Vieira

Academic Editor

PLOS ONE

Journal Requirements:

Reviewers' comments:

Reviewer's Responses to Questions

**Comments to the Author**

1. If the authors have adequately addressed your comments raised in a previous round of review and you feel that this manuscript is now acceptable for publication, you may indicate that here to bypass the “Comments to the Author” section, enter your conflict of interest statement in the “Confidential to Editor” section, and submit your "Accept" recommendation.

Reviewer #3: All comments have been addressed

Reviewer #4: (No Response)

Reviewer #5: (No Response)

2. Is the manuscript technically sound, and do the data support the conclusions?

Reviewer #3: Yes

Reviewer #4: Partly

Reviewer #5: Yes

3. Has the statistical analysis been performed appropriately and rigorously? 

Reviewer #3: I Don't Know

Reviewer #4: Yes

Reviewer #5: Yes

4. Have the authors made all data underlying the findings in their manuscript fully available?

Reviewer #3: Yes

Reviewer #4: Yes

Reviewer #5: Yes

5. Is the manuscript presented in an intelligible fashion and written in standard English?

Reviewer #3: Yes

Reviewer #4: Yes

Reviewer #5: Yes

6. Review Comments to the Author

Reviewer #3: The findings in the study highlight the complexity of developing RIs . the results suggest that racial/ethnic subpopulations have unique distributions in laboratory tests, accordingly further work is needed to establish of racial/ethnic-specific RIs that may have significant clinical and public health implication for more accurate disease diagnosis and appropriate treatment to improve quality of patient care. ** the author should give a more emphasizes in the recommendation

I hope the study may draws attention to a possible revision of the WHO reference intervals in regard

I think the manuscript is going to be useful like that

no further comments

Reviewer #4: Ruzhanskaya et al present complete blood count and iron related marker reference intervals (RIs)from subjectively healthy Russians recruited as part of the global study on reference intervals carried out by the Interbational Federation of Clinical Chemistry. They use both parametric and non-parametric approaches to derive RIs after secondarily excluding individuals based on different criteria including latent abnormal value exclusion. Overall, this is a well written paper presenting useful evidence based information on CBC reference intervals for adoption in Russian.

Major concerns

1. The statistical approach is subjective and not applied in a consistent manner that would make the results reproducible. For example, the basis for excluding ‘outliers’ appears to be arbitrary since the RI width for both upper and lower limits was increased by 3% for no other reason other than to reduce number of reference individuals excluded.

2. Exclusion of individuals was partly based on manufacturer recommended cut-offs for ferritin (notably higher than WHO recommended cut-offs) despite not knowing whether they are appropriate for the Russian population. It is possible that many individuals were inappropriately excluded as being iron deficient. As observed, the derived RIs for ferritin in women had a lower limit that was lower than the manufacturer recommend cut-off.

3. It is indicated that an SDR>0.4 was part of the criteria used to determine the need for partitioning. However, partitioning based on sex was done for platelets despite having an SDRsex that was less than 0.4.

4. Given the complexity of the analyses, it would be beneficial to have a clear schema that summarizes the logic behind partitioning of the RIs. For example, for the bias ratio, was 0.57 or 0.375 used as the criteria? It is not clear the basis for multiplying 0.375 with the square root of two. What does the square root of two represent.

Minor concerns

See in-text comments for minor comments.

Reviewer #5: The article plays an important role in standardizing reference values for a population with little ethnic diversity and demonstrating variations and correlations between reference values, influenced by sex, age and BMI. However, I suggest that the authors provide more details about the secondary patient exclusion criteria (information about health status acquired in the questionnaire). It was not clear how the exclusion and inclusion, even presenting a reference to Cohen, required more details.

7. PLOS authors have the option to publish the peer review history of their article (what does this mean?). If published, this will include your full peer review and any attached files.

Reviewer #3: **Yes: **ANISA H. ALBITI

Reviewer #4: No

Reviewer #5: No

---

## [Author Response · Author response to Decision Letter 1]

23 Apr 2024

Reviewer #3: The findings in the study highlight the complexity of developing RIs. the results suggest that racial/ethnic subpopulations have unique distributions in laboratory tests, accordingly further work is needed to establish of racial/ethnic-specific RIs that may have significant clinical and public health implication for more accurate disease diagnosis and appropriate treatment to improve quality of patient care. ** the author should give a more emphasizes in the recommendation

I hope the study may draws attention to a possible revision of the WHO reference intervals in regard

I think the manuscript is going to be useful like that

no further comments

Our response → We appreciate a great deal for reviewing our manuscript. We are pleased to have your encouraging message to our work. Yes, it was a very complex and demanding work to determine RIs appropriately for many heterogenous analytes simultaneously. We highlighted the point in the end of Introduction, in the beginning of the Discussion, and in the Conclusion. We hope the table we made for international comparison of recently reported RIs is of relevance to the readers.

Reviewer #4: Ruzhanskaya et al present complete blood count and iron related marker reference intervals (RIs)from subjectively healthy Russians recruited as part of the global study on reference intervals carried out by the Interbational Federation of Clinical Chemistry. They use both parametric and non-parametric approaches to derive RIs after secondarily excluding individuals based on different criteria including latent abnormal value exclusion. Overall, this is a well written paper presenting useful evidence based information on CBC reference intervals for adoption in Russian.

Our response →We are very grateful for your time and efforts in reviewing our manuscript. We are pleased to have your positive over-all evaluation to our work. Besides, we appreciate a great deal for providing us with invaluable comments to improve the scientific quality of our manuscript. We addressed the issues raised and corrected one by one as shown below.

Major concerns

1. The statistical approach is subjective and not applied in a consistent manner that would make the results reproducible. For example, the basis for excluding ‘outliers’ appears to be arbitrary since the RI width for both upper and lower limits was increased by 3% for no other reason other than to reduce number of reference individuals excluded.

Our response → We appreciate for pointing out the importance of reproducibility in our methodologies. We admit the problem of somewhat arbitrary nature of our condition setting to perform the latent abnormal value exclusion (LAVE) method. In fact, there are many options to choose from: i.e., (1) what analytes for use as reference tests (RTs: 5, 7, or more), (2) how many times to adopt for iterative adjustment of RIs (i.e., usually 6 times but can be 5 or 7~), (3) when to adjust the RIs of RTs (i.e., no adjustment of the RI, 3% or 5% extension on both-ends of the RI), etc. Therefore, these adjustments have been done in a trial-and-error fashion over the past 25 years, and we obtained empirical thresholds of 6-time repetitive adjustment of RIs and 0~5% extension of RIs for RTs were most suitable in terms of “how many data remain after LAVE procedure” and “how-much narrowing of the RI occurs”. 

In any case, we added the following point in the Limitation as to the need for optimizing the performance of the LAVE method by adjusting the settings.

“Another limitation of this study is somewhat arbitrary nature of setting conditions for secondary exclusion using the LAVE method. In fact, there can be many options to attain optimal LAVE effect without much data-size reduction, which depend on how many reference tests to set up and how much to expand the RI limits in judging test result abnormality. Although we adopted 3% expansion on both-end of the RI from our past trial and error experiences, other settings can lead to similar results.”

2. Exclusion of individuals was partly based on manufacturer recommended cut-offs for ferritin (notably higher than WHO recommended cut-offs) despite not knowing whether they are appropriate for the Russian population. It is possible that many individuals were inappropriately excluded as being iron deficient. As observed, the derived RIs for ferritin in women had a lower limit that was lower than the manufacturer recommend cut-off.

Our response → We understand the reviewer’s concern over the use of unconfirmed ferritin (Ferr) RI to select women with iron-deficiency anemia (IDA). Unfortunately, our writing seems to have aroused some confusion. In our final determination of RIs, we did not apply manufacturer’s Ferr cutoff value in excluding individual with latent anemia. Rather, we just needed to choose a subgroup of women without IDA by the levels of Ferr and iron in order to demonstrate the effectiveness of the LAVE method in excluding the influence of IDA (please refer to Fig. 3). Therefore, it is certainly unlikely that “many individuals were inappropriately excluded as being iron deficient”.

In any case, as for the manufacturer recommended cutoffs for female ferritin, we understand that it was fully aligned to those provided in Russian guidelines of National Hematology Laboratory Society (NHLS; 11 mcg/L), which was close to WHO cutoff (15 mcg/L).

Regarding our finding that “the derived RIs for ferritin in women had a lower limit that was lower than the manufacturer recommend cut-off”. We must acknowledge it as a fact for Russian women despite our best efforts of reducing the influence of IDA by using the LAVE methods. We recognized this point as we wrote about it in the Discussion as pasted below:

“In Russian females we observed a slightly reduced LL of Hb in comparison to the AGA and RNHS guidelines (114 vs. 120 g/L) [36,37], meanwhile significantly lower ferritin LL was observed in both age groups <45 and ≥45 years (5.5 and 5.3 vs. 45 and 11 µg/L, respectively) [36-38]. Even a lower level of ferritin was reported in the Turkish and Japanese studies (4.3 and 3 µg/L, respectively) [5,6]. Low levels of ferritin in age group <45 years are associated with reduced iron storage due to menstrual bleeding, pregnancy or latent anemia [8]. Bawua et. al. showed age-related increase in ferritin and decrease in TFR, which was confirmed in this study. However, LL for ferritin in Russian females for the age group >45 is still very low (5.3 µg/L).”

3. It is indicated that an SDR>0.4 was part of the criteria used to determine the need for partitioning. However, partitioning based on sex was done for platelets despite having an SDRsex that was less than 0.4.

Our response → We applied two tiers of criteria in judging the need for partitioning reference values (RVs): The first by using (1) SD ratio (SDR): between-group SD divided by within-group SD, then (2) by using bias ratio (BR) at lower and upper limits (LL, UL) of the RI

Accordingly, there can be situation where SDRsex is less than the threshold (<0.40), but actual BR at UL exceeds its threshold of 0.57. Such discrepancy was typically observed in RVs of platelets and Eos as picked-up below.

4. Given the complexity of the analyses, it would be beneficial to have a clear schema that summarizes the logic behind partitioning of the RIs. For example, for the bias ratio, was 0.57 or 0.375 used as the criteria? It is not clear the basis for multiplying 0.375 with the square root of two. What does the square root of two represent.

Our response → We are sorry about the complicate nature of defining threshold for judging the magnitude of “between-group” differences. The important distinction between SDR and BR is that SD is calculated as standard deviation (√(∑_(i=1)^n▒〖(x_i-x ®)〗^2 /(n-1)) while BR is a calculated as a difference of two values |x ®_1-x ®_2 | or bias. 

Therefore, to set the threshold of BR aligned to the threshold of SDR=0.4 (for judging between-sex difference), we multiplied SDR=0.4 by √2 to obtain its threshold of 0.4×√2=0.57: i.e., absolute difference (bias) of two values (x1 and x2), |x_1-x_2 | is √2 times larger than SD of the two values

√({(x_1-x ® )^2+(x_2-x ® )^2 }/(2-1) ) . Please refer to the proof below in brown texts for cases of SDR of two groups with n1 and n2.

Another complicate thing was our use of two different thresholds (1) for judging between-sex differences and (2) for judging the effect of LAVE method. The former (SDR=0.4 or BR=0.57) was inferred from conventional clinical practice of setting sex-specific, while for the latter, the threshold of BR=0.375 was decided in analogy to the conventional allowable analytical bias.

In the revised manuscript, we added the following red-texts to clarify our scheme of using different threshold. 

“4.2) Criteria for partitioning RVs 

The standard deviation ratio (SDR) was used as a criterion for assessing the need of partitioning RVs by sex and age. To calculate SDR, pure components of between-sex SD (SDsex) and between-age SD (SDage) were first computed by two-level nested ANOVA and within-group SD (approximately 1/4 the width of RI, or SDRI). The SDRs for sex or age (SDRsex, SDRage) were calculated by taking respective ratios of SDsex and SDage to SDRI. Since SDRage often differs between sexes, it was also calculated by one-way ANOVA after partitioning RVs by male and female, as SDRageM, SDRageF. The need for partition of RVs was considered by setting SDR>0.4 as a primary guide: i.e., the threshold of 0.4 was inferred from conventional clinical practice of setting sex-specific RIs [14]. 

However, SDR can be too sensitive, when the width of RI constituting the denominator of SDR is narrow. Conversely, SDR may be insensitive, when between-subgroup differences occur only at the periphery of distribution (LL or UL) — SDR represents between-subgroup differences at the center of the distributions. Therefore, we additionally considered actual difference (bias) at LL or UL as “bias ratio” (BR) using the following formula, illustrating a case of gender difference:

 BRLL = (|〖LL〗_M－〖LL〗_F |)/((〖UL〗_MF－〖LL〗_MF)/3.92), BRUL = (|〖UL〗_M－〖UL〗_F |)/((〖UL〗_MF－〖LL〗_MF)/3.92) 

where subscript M, F, and MF represent male, female, and male+female, respectively. The denominator of each formula represents the standard deviation (SDRI) comprising the RI, the width of which corresponds to 3.92 times SDRI. To make the threshold of BR aligned to SDR with the common denominator (SDRI), we multiplied the threshold of SDR=0.4 by √2 to obtain BR’s threshold of 0.57 (i.e., absolute difference (bias) of two values (x1 and x2), or |x1−x2|, is √2 times larger than SD of the two values).

 Whereas, when BR was used for judging the effectiveness of the LAVE method, its threshold was set according to the conventional specification of allowable analytical bias (=difference) of a minimum level: 0.375 ×√(〖SD〗_G^2+〖SD〗_I^2 ) (= SDRI) [17]. Hence 0.375 was set as the threshold for BR in contrast to the above-mentioned threshold of BR for between-sex bias, which was inferred from conventional clinical practice.

Minor concerns

See in-text comments for minor comments.

Our response → We requested the editorial office for the “in-text comments”, but there was no response.

Reviewer #5: The article plays an important role in standardizing reference values for a population with little ethnic diversity and demonstrating variations and correlations between reference values, influenced by sex, age and BMI. However, I suggest that the authors provide more details about the secondary patient exclusion criteria (information about health status acquired in the questionnaire). It was not clear how the exclusion and inclusion, even presenting a reference to Cohen, required more details.

Our response →We are very grateful to your time and efforts for reviewing our manuscript and providing us with comments to make our manuscript clearer for the readers to follow.

Regarding the inclusion and exclusion criteria, we wrote it clear in the beginning of the Materials and Methods as pasted below:

“Materials and Methods 

1) The study protocol and recruitment 

Healthy volunteers were recruited with the objectives of determining RIs for hematological and related parameters and exploring the source of variations of each analyte for better derivation and clinical interpretation of test results. The study protocol was prepared in harmonization to that provided by the IFCC, C-RIDL [3]. The inclusion and exclusion criteria adopted in the recruitment were as follows: 

Inclusion criteria 

The participants were feeling well, between the ages of 18 and 65 and ideally, they were not taking any medication or supplements. However, if they were taking medication, doses and frequency were recorded accordingly. 

Exclusion criteria 

The participants were excluded if any of the following were applicable: if he or she was diabetic and on oral therapy or insulin, had a history of chronic liver or kidney disease, had results from their blood samples that clearly point to a severe disease, had been a hospital in-patient or been subjectively seriously ill during the previous 4 weeks of participation, donated blood in the previous 3 months, if they were a known carrier of HBV, HCV, or HIV, if she was pregnant or within one year after childbirth and if they had participated in another research study involving an investigational product in the past 12 weeks. 

This rather lenient criteria clearly entailed the need for secondary exclusion after measurements based on the pattern of the actual test results as described below. The health-status information acquired from the questionnaire (i.e., BMI, habits of regular exercise, drinking, or smoking, etc. [3]) was also used to consider for the secondary exclusion.”

Regarding the ways for secondary exclusion, we did it meticulously because of its necessity as described above in the italic part of descriptions. The most important scheme of secondary exclusion was (1) to exclude highly prevalent latent anemia using the latent abnormal values exclusion method, and (2) to exclude obese individuals for determining RIs for analytes with significant association of their values with BMI. As for the effect of smoking or allergy, we found their effects were negligible by the multiple regression analysis. Since these descriptions are already in the current manuscript, we hope it is not necessary to add more detailed descriptions.

As for the Cohen’s criteria, it is customary to judge the magnitude of observed correlations (partial correlation coefficients) in terms of “effect size”, rather than judging the magnitude by P value, which depends on data size and may not represent practically important degree of “correlation”.

Therefore, we modified its description in Statistical methods as follows:

“Since statistical testing of rp is too sensitive with large data size, we interpreted its practical significance (“effect size”) in reference to the Cohen’s guide [16] as small 0.1< |rp| <0.3; medium 0.3< |rp| <0.5, large |rp| < 0.5. Hence, we regarded |0.3| <rp as significant when considering the influence of a given factor on the RVs.”

----------------

We hope our answer to the Reviewer-5’s comments are satisfactory also to answer to the following comments kindly offered by the Academic Editor: 

ACADEMIC EDITOR: 

Dr. Ichihara

This is an interesting manuscript that show relevant information on reference values of some biochemical parameters. I recommend to evaluate the suggestion of the reviewer 5 "I suggest that the authors provide more details about the secondary patient exclusion criteria (information about health status acquired in the questionnaire). It was not clear how the exclusion and inclusion, even presenting a reference to Cohen, required more details". Such information is relevant to strengten your results. Our decision is justified on PLOS ONE’s publication criteria and not, for example, on novelty or perceived impact.

---

## [Editor Report · Decision Letter 2]

6 May 2024

Derivation of Russian-specific reference intervals for complete blood count, iron markers and related vitamins

PONE-D-23-22704R2

Dear Dr. Kiyoshi Ichihara

We’re pleased to inform you that your manuscript has been judged scientifically suitable for publication and will be formally accepted for publication once it meets all outstanding technical requirements.

Kind regards,

José Luiz Fernandes Vieira

Academic Editor

PLOS ONE

---

## [Editor Report · Acceptance letter]

14 Jun 2024

PONE-D-23-22704R2 

PLOS ONE

Dear Dr. Ichihara, 

I'm pleased to inform you that your manuscript has been deemed suitable for publication in PLOS ONE. Congratulations! Your manuscript is now being handed over to our production team.

Kind regards, 

on behalf of

Dr. José Luiz Fernandes Vieira 

Academic Editor

PLOS ONE